

**Mg/Ca and $\delta^{18}$O in multiple species of planktonic foraminifera from 15 Ma to Recent**
Flavia Boscolo-Galazzo[1*], David Evans[2], Elaine M. Mawbey[3], William R. Gray[4], Paul N.
Pearson[3,5], Bridget S. Wade[3]
[1]Bremen University, MARUM, Center for Marine Environmental Sciences (Germany);
[2]School of Ocean and Earth Science, University of Southampton, European Way, SO14 3ZH,
Southampton (UK);
[3]Department of Earth Sciences, University College London, London (UK);
[4]Laboratoire des Sciences du Climat et de l'Environnement (LSCE/IPSL), Université Paris-
Saclay, Gif-sur-Yvette (France).
[5]School of Earth and Environmental Sciences, Cardiff University, Cardiff (UK)
*Corresponding author: fboscologalazzo@marum.de
D.evans@soton.ac.uk
mawbeye@gmail.com
william.gray@lsce.ipsl.fr
p.pearson@ucl.ac.uk
b.wade@ucl.ac.uk
**Abstract**
The ratio of the trace element Mg over Ca (Mg/Ca) and the oxygen isotopic composition ($\delta^{18}$O) of
foraminiferal calcite are widely employed for reconstructing past ocean temperatures, although
geochemical signals are also influenced by several other factors that vary temporally and spatially.
Here, we analyze a global dataset of Mg/Ca and $\delta^{18}$O data of 59 middle Miocene to Holocene
species of planktonic foraminifera from a wide range of depth habitats, many of which have never
been analyzed before for Mg/Ca. We investigate the extent to which Mg/Ca and $\delta^{18}$O covary





through time and space, and identify several sources of mismatch between the two proxies. Once
the data are adjusted for long term non-thermal factors, Mg/Ca and $\delta^{18}$O are overall positively
correlated in a way consistent with temperature being the dominant controller of both through
space and time and across many different species, including deep-dwellers. However, we identify
several species with systematic offsets in Mg/Ca values, to which multispecies calibrations should
be applied with caution. We can track the appearance of such offsets through ancestor-descendent
species over the last 15 million years and propose that the emergence of these offsets may be the
geochemical expression of evolutionary innovations. We find virtually all of the Mg/Ca and $\delta^{18}$O-
derived temperatures from the commonly used genera *Globigerinoides* and *Trilobatus* are within
uncertainty of each other, highlighting the utility of these species for paleoceanographic
reconstructions. Our results highlight the potential of leveraging information from species lineages
to improve sea surface temperature reconstruction from planktonic foraminifera over the
Cenozoic.
**1. Introduction**
Geochemical analyses of foraminifera are commonly applied to reconstruct paleoceanographic
conditions, such as marine temperatures, and therefore infer past climatic changes. In particular,
the fossil tests of planktonic foraminifera (calcareous zooplankton) provide one of the most widely
used paleoclimate archives. Here we focus on two of these parameters: $\delta^{18}$O and Mg/Ca, both of
which have been used widely as temperature proxies.
The oldest and possibly most widely utilized of these proxies is the ratio of oxygen isotopes in
their calcite test which, due to slight differences in reactivity of molecules containing the different
isotopes, is temperature-dependent (Urey, 1947; see Pearson, 2012 for review). This effect has
been quantified in experiments with inorganic calcite (e.g., Kim and O'Neill, 1997) and planktonic



foraminifera in culture (e.g., Erez and Luz, 1983; Bemis et al., 1998). Tests of planktonic
foraminifera calcifying in warmer waters are depleted in $^{18}$O relative to species living in cooler
waters (Emiliani, 1954). A second, more recently established paleoclimate proxy is the ratio of
magnesium to calcium in test calcite (Chave 1954; Nürnberg et al., 1996). During inorganic
precipitation experiments, the Mg/Ca ratios of calcite were found to be higher at greater
temperatures (Mucci, 1987). This relationship led to the in-depth exploration of Mg/Ca ratios in
planktonic and benthic foraminifera and its potential application as a temperature proxy through
culturing (Lea et al., 1999; von Langen et al., 2005), core top (Nürnberg, 1995; Elderfield and
Gassen, 2000) and sediment trap studies (Anand et al., 2003).
As they represent two different chemical systems, the Mg/Ca and oxygen stable isotope ratios in
foraminifera are often used together as independent temperature proxies. For instance, $\delta^{18}$O
derived calcification temperatures have been combined with Mg/Ca data to derive Mg/Ca
temperature calibrations (e.g., Anand et al., 2003; McConnel and Thunell, 2005; Mohtadi et al.,
2009). Other studies have applied these two systems together to infer the influence of
environmental parameters such as seawater salinity on Mg/Ca (e.g., Mathien-Blard and Bassinot,
2009; Hönisch et al., 2013) and global ice volume (e.g., Lear et al., 2000; Katz et al., 2008). Works
such as these assume covariance of the two proxies for any given sample, which should be the case
if both systems are impacted purely by calcification temperature. Nonetheless, there are known
non-thermal effects influencing both Mg/Ca and $\delta^{18}$O. For oxygen isotope values, these include
the oxygen isotopic composition of seawater ($\delta^{18}$O$_{sw}$) and to a lesser degree, seawater pH or
carbonate ion concentration (Spero et al., 1997; Zeebe, 1999). Seawater carbonate chemistry has
also been shown to impact the Mg/Ca proxy. Culture and sediment trap studies demonstrate surface
ocean seawater pH can influence Mg/Ca in planktonic foraminifera (Lea et al 1999; Evans et al.,





2016a; Gray et al 2018), with the sensitivity of Mg/Ca to pH appearing to vary between species
(Gray and Evans 2019). Mg/Ca values of foraminifera are also dependent on the Mg/Ca of
seawater (Evans et al., 2016b), and both oxygen isotope and Mg/Ca values can be impacted by test
recrystallization (Dekens et al., 2002). Mg/Ca values are susceptible to the preferential loss of Mg
during dissolution, and are thus influenced by the calcite saturation state of bottom waters
(Regenberg et al 2014; Tierney et al 2019). Seawater salinity has a minor secondary effect on
Mg/Ca values (Kisakürek et al., 2008, Hönisch et al., 2013) and whilst salinity has little direct
effect on oxygen isotopes, a change in salinity is usually accompanied by a change in $\delta^{18}O_{sw}$
because hydrological processes such as evaporation and precipitation are closely coupled
(LeGrande and Schmidt 2006). Lastly, so-called 'vital effects', which lump together a wide variety
of species-specific processes such as metabolism (including the process of calcification and the
incorporation of metabolic products), the position within the water column and life cycle depth
migration, the presence of photosymbionts, and seasonality (see summary in Schiebel and
Hemleben, 2017), also add complexity to the interpretation of both the oxygen isotope and Mg/Ca
proxies.
Here we use the dataset published in Boscolo-Galazzo, Crichton et al., (2021), to examine
covariance between Mg/Ca and $\delta^{18}O$ in planktonic foraminifera extracted from sediments across
a wide range of geographic locations, time intervals, and species. The dataset is composed of $\delta^{18}O$
and Mg/Ca data measured on 59 species of planktonic foraminifera, of which 24 have never before
been measured for Mg/Ca (Supplementary Tables 1, 2). The data are from different ocean basins
and latitudes and a range of ages between the middle Miocene (~15 million years ago, Ma) and
the Holocene. Paired Mg/Ca and $\delta^{18}O$ were measured on the same samples, hence this dataset is
ideally suited to isolate potential ecological, environmental and preservational factors which may





imprint Mg/Ca or $\delta^{18}O$ or both, and which are otherwise impossible to recognize in studies
focusing on a limited number of species, a narrow study area or time interval. In particular, it
provides the unique opportunity to simultaneously: (1) compare coupled $\delta^{18}O$ and Mg/Ca data on
a broader than usual geographical and temporal scale; (2) Compare coupled $\delta^{18}O$ and Mg/Ca data
across species of different ecologies; (3) Evaluate Mg/Ca data of extinct species against those of
their modern descendants; (4) Test whether temperature can still be recognized as predominantly
driving covariance in the dataset when spatial, temporal and ecological variables are
simultaneously in play.
**2. Material and Analytical methods**
2.1 Material
The dataset (Boscolo-Galazzo, Crichton et al., 2021) was produced from a range of globally and
latitudinally distributed DSDP (Deep Sea Drilling Program), ODP (Ocean Drilling Program), and
IODP (Integrated Ocean Drilling Program/International Ocean Discovery Program) sites (Fig. 1)
which are high in carbonate and composed of calcareous nannofossils and foraminiferal pelagic
oozes, with some input of siliceous plankton. Sites were selected based on the best available global
and temporal coverage and preservation of foraminifera. Planktonic foraminiferal preservation
ranges from excellent to very good (recrystallized but lacking overgrowth and infilling) (Boscolo-
Galazzo, Crichton et al., 2021) with the exception of Sites U1490 and U1489, where there is some
overgrowth and infilling in the middle Miocene (Fayolle and Wade, 2020; Boscolo-Galazzo,
Crichton et al., 2021). The target time intervals selected for sampling were 0, 2.5, 4.5, 7.5, 10, 12.5
and 15 Ma. Biostratigraphic analysis was used to assess age using the biochronology of Wade et
al. (2011) calibrated to the time scale of Lourens et al. (2004) (Supplementary Table 1).



2.2 Planktonic foraminifera
Fifty-nine species of planktonic foraminifera were analysed for Mg/Ca and $\delta^{18}$O. Planktonic
foraminiferal were picked from three constrained size fractions: 180-250 μm, 250-300 μm and
300-355 μm. Planktonic foraminiferal geochemistry can change through size (e.g., Birch et al.,
2013), so here we used data from the size fraction 250-355 μm only, giving a total of 57 species
in the dataset. For abundant species, up to 80 specimens were picked for geochemical analysis,
with as many as possible picked in the case of less common species. Hence, our foraminiferal data
represent an average from multiple specimens. Paleodepth habitat attributions follow Boscolo-
Galazzo, Crichton et al. (2021) and Boscolo-Galazzo et al. (2022). Planktonic foraminiferal
taxonomy follows the concepts described in Boscolo-Galazzo et al. (2022).

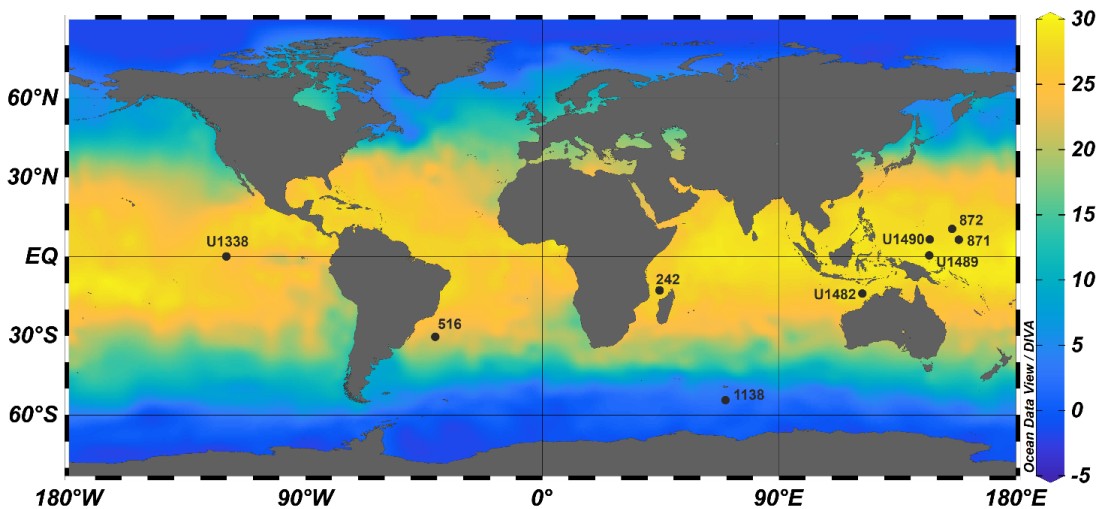


**Figure 1.** Site map with present-day mean annual sea surface temperatures (°C) from the World
Ocean Atlas 2013 (Locarnini et al., 2013).






2.3 Trace element and stable isotope analysis
Picked planktonic foraminifera were crushed between two glass slides to open all large chambers.
When there was enough material, the crushed sample was split for stable isotope and trace element
analysis. The trace element split was cleaned using a protocol to remove clays and organic matter
(step A1.1-A1.3 of Barker et al. (2003)). The samples did not undergo reductive cleaning due to
their fragility and small sample size, and because the reductive step may cause preferential removal
of high Mg/Ca calcite from the test (Yu et al., 2007). Samples were dissolved in trace metal pure
0.065 M $HNO_3$, then diluted with trace metal pure 0.5 M $HNO_3$ and analysed at Cardiff University
on a Thermo Fisher Scientific Element XR ICP-MS against standards with matched calcium
concentration to reduce matrix effects (Lear et al., 2002). Long term analytical precision
determined from consistency standards (CS1 and CS2) with Mg/Ca ratios of 1.24 mmol/mol and
7.15 mmol/mol are ~0.7 and ~0.8% (relative standard deviation). Mg/Ca was plotted against Fe/Ca
and Mn/Ca to assess whether there was any relationship as a result of the presence of Fe-Mn
oxyhydroxides affecting Mg/Ca, but there was no correlation between the contaminant indicators
and Mg/Ca (Supp. Fig. 1).
Stable isotopes were measured on a Delta V Advantage with Gasbench II mass spectrometer at the
Cardiff University stable isotope facility. Stable isotope results were calibrated to the VPDB scale
using an in-house carbonate standard (Carrara marble). Analytical precision was 0.05‰ for $\delta^{18}O$
and 0.05‰ for $\delta^{13}C$.
2.4 Data analysis





Before performing the analysis, we screened the dataset for outliers, and removed one anomalously
high datapoint with a Mg/Ca value >9 mmol/mol which we attributed to analytical error
(Supplementary Table 1).
2.4.1 Formulation of theoretical relationships between Mg/Ca and $\delta^{18}O$
To test for covariation between Mg/Ca and oxygen isotope data, we regressed the data against each
other and compared the observed relationship with that expected from modern calibrations. We
did this to initially explore the dataset and what kind of relationship we might expect between
Mg/Ca and $\delta^{18}O$ and whether this manifests in the dataset, before applying corrections for the non-
thermal influences on both proxies. Given the complexity of the sample set (e.g., multiple species,
ages, locations, preservation), different expected relationships between Mg/Ca and $\delta^{18}O$ are
possible, which depend on: i) species-specific vital effects, ii) the non-thermal controls on Mg/Ca,
(salinity, pH, Mg/Ca$_{sw}$), iii) non-thermal controls on $\delta^{18}O$ (pH/[$CO_3^{2-}$]), $\delta^{18}O_{sw}$, as well as how
these factors change through time. To account for this, we calculated a number of possible
expected theoretical relationships to give a sense of how much of the scatter in the raw data is
likely to be explicable by these factors and inform our following data-analysis accordingly. We
stress that this exercise was conducted as a mean of exploring the whole data set; no single
relationship will be able to explain the dataset because it is influenced by multiple, often
interlinked, variables.
Expected theoretical relationships were calculated starting with modern laboratory culture
calibrations, onto which the key non-thermal long-term and spatial controls on these proxies were
sequentially added to demonstrate how much each of these is expected to shift the slope of the
expected Mg/Ca-$\delta^{18}O$ relationship (Fig. 2A). Specifically, we i) combined the calibrations for
*Globigerinoides ruber* and *Trilobatus sacculifer* of Gray & Evans (2019) with the $\delta^{18}O$-



temperature equation of Erez & Luz (1983), ii) added the impact of a 0.15 unit whole ocean pH
change (approximating the magnitude of the Neogene whole ocean change, e.g., Rae et al., 2021)
using the pH-Mg/Ca slope for *G. ruber* as an example (note that this is only applicable to species
that show a pH sensitivity) (Evans et al., 2016a), iii) included the expected control of temperature
on pH via the T-dependent dissociation of water ($K_W$), i.e., temperature-driven pH changes within
a given time interval independent of whole ocean pH shifts (Gray et al., 2018), iv) showed the
impact of Mg/Ca$_{sw}$ half of the modern ratio (Evans et al., 2016b), v) included the effect of pH or
[$CO_3^{2-}$] on $\delta^{18}O$ (Spero et al., 1997; Zeebe, 1999) given the covariance of temperature and pH
described in point iii above using the multispecies average slope of Gaskell et al. (2023), and
finally vi) explored the likely impact of the covariance of $\delta^{18}O_{sw}$ and temperature that is
characteristic of the modern ocean and arises from the broad coupling of the hydrological cycle
with surface temperatures. Specifically, this latter influence was calculated by combing SST data
from the 2013 World Ocean Atlas (Locarnini et al., 2013) and $\delta^{18}O_{sw}$ from LeGrande & Schmidt
(2006), taking all surface ocean data except that from polar meltwater regions, which demonstrates
that, on average, in the modern ocean $\delta^{18}O_{sw}$ increases by 0.0425 ‰ per °C SST increase.
Each of these factors was applied additively such that (e.g.) the fourth factor listed above
(Mg/Ca$_{sw}$) in Fig. 2 includes numbers 1 through 3. The sum of the influence of these factors on
the theoretical $\delta^{18}O$-Mg/Ca relationship is represented by the thick blue line in Figure 2A and the
black line in Figures 3, 6, 7 and 8, which has a slope of –2.08 in $\delta^{18}O$-ln(Mg/Ca) space.
The magnitude of some of these potential non-thermal controls on the two proxies over the time
interval studied here are reasonably well constrained. Specifically, the long-term whole ocean pH
and Mg/Ca$_{sw}$ changes are sufficiently well known (Rae et al., 2021; Zhou et al., 2021; Brennan et
al., 2013) that they can be "subtracted" out of the raw proxy values, given that they are likely to



apply to all or most species in the dataset. As such, we next explored the degree to which the
Mg/Ca-$\delta^{18}$O covariation improves once long-term whole ocean pH and Mg/Ca$_{sw}$ changes are
removed. To avoid (possibly incorrect) a-priori assumptions regarding, for example, which
Mg/Ca-temperature calibration should be applied to each species in the dataset and the degree to
which surface ocean $\delta^{18}$O$_{sw}$ has varied at the study sites, we  initially did this  keeping the Mg/Ca
and $\delta^{18}$O comparison in raw proxy space and: 1) converted the raw Mg/Ca values to temperature
using the multispecies Mg/Ca-temperature calibration from Gray and Evans (2019), together with
our best estimate of pH and Mg/Ca$_{sw}$ (as described below (§2.4.2)), and 2) converted the
temperatures back into Mg/Ca using the same calibration but modern seawater Mg/Ca and pH. In
addition, we subtracted out the long-term whole ocean change in $\delta^{18}$O$_{sw}$ related to continental ice
growth using the sea level curve of Rohling et al. (2021) and a sea level-$\delta^{18}$O$_{sw}$ scaling factor of
1‰ per 67 m. This results in a raw proxy dataset in which the aforementioned long-term non-
thermal factors are no longer present and which can be used to evaluate the occurrence of residual
scatter independent of the long term non-thermal controls on Mg/Ca and $\delta^{18}$O (Fig. 2B).
2.4.2 Transformation of proxy values into paleotemperature
Measured foraminifera Mg/Ca was transformed into paleotemperature using the *MgCaRB* tool
(Gray       &       Evans,       2019;       https://github.com/willyrgray/MgCaRB       (*R*);
https://github.com/dbjevans/MgCaRB (*Matlab*)) which takes into account:
-    Salinity. Although this has a minor effect on Mg/Ca (Hönisch et al., 2013), whole ocean

changes are nonetheless accounted for using a salinity reconstruction derived from scaling

the $\delta^{18}$O benthic stack (Westerhold et al., 2020) to the sea level record of Spratt & Lisiecki

(2016) back to 8 Ma, before which that of Miller et al. (2005) was used, rescaled to match

the $\delta^{18}$O-derived reconstruction at 8 Ma and a sea level of +67 m in an ice-free world at 50





Ma. We applied the multispecies salinity sensitivity of Gray & Evans (2019) to all species
(3.6% per salinity unit).
-    pH. Long-term whole ocean changes were derived from a smoothing spline fit to the boron
isotope-derived pH data compiled by Rae et al. (2021). We applied species-specific pH-
Mg/Ca sensitivities of Gray & Evans (2019) where available for a given species/lineage
(discussed in more detail below) and used the multispecies sensitivity in all other cases.
-    Mg/Ca$_{sw}$ was derived by combining the [Ca$^{2+}$$_{sw}$] record of Zhou et al. (2021) with a
smoothing spline fit to the fluid inclusion [Mg$^{2+}$$_{sw}$] data given in Brennan et al. (2013).
Raw Mg/Ca values were adjusted using the equation Mg/Ca$_{corrected}$ = Mg/Ca$_{raw}$ $\times$
Mg/Ca$_{sw}$$^{H}$/5.2$^{H}$, where H = 0.64 based on a data compilation of three foraminifera species
and inorganic calcite (Holland et al., 2020; Evans et al., 2015; 2016b; Mucci & Morse,

1983).

Because the dataset includes a mix of extant and extinct species, some of these never measured for
trace elements before (Supplementary Table 2), or lacking an extant/well-calibrated modern
relative, when converting Mg/Ca to temperature we started by applying a multispecies equation,
as is typically done for extinct species. Specifically, we used the multispecies Mg/Ca-temperature
equation of Gray & Evans (2019) and applied the multispecies pH, salinity, and temperature
sensitivities, together with the *Globigerinoides ruber* exponential coefficient as most of the species
for which high quality data exist are known to be characterized by a Mg/Ca-pH sensitivity (Lea et
al., 1999; Kisakürek et al., 2008; Evans et al., 2016a). We subsequently applied species-specific
calibrations to selected lineages to explore the degree to which scatter in the dataset can be
accounted for by taking into account phylogenetic relationships among ancestor-descendent
species. Specifically, the *Trilobatus sacculifer* calibration was applied to the *Trilobatus trilobus* -



*Trilobatus sacculifer* lineage, and the *Orbulina universa* calibration was applied to the
*Preaorbulina-Orbulina* lineage, both from laboratory culture studies following Gray & Evans
(2019).   To *Neogloboquadrina* and its descendent lineage *Pulleniatina* we applied a
*Neogloboquadrina pachyderma* calibration with the sensitivies of Tierney et al. (2019)
(implemented with a re-fit to the dataset following the *MgCaRB* approach). We then evaluate the
improvement relative to the multispecies calibration in samples spanning the middle Miocene to
modern. The attribution of phylogenetic relationships follows Aze et al. (2011), Spezzaferri et al.
(2018), Leckie et al. (2018) and Fabbrini et al. (2021).  All uncertainties were fully propagated via
Monte Carlo simulation, including those related to: analysis, calibration coefficients, and the 95%
confidence intervals on the salinity, pH, and $Mg/Ca_{sw}$ reconstructions, with $10^4$ random draws of
each within the uncertainty bounds used to generate the reported values and 95% CI (2.5th, 50th,
and 97.5th percentiles of the resulting dataset).
The conversion of $\delta^{18}O$ to paleotemperature followed Gaskell et al. (2023) using: the bayfox
calibration (Malevich et al., 2019) and the global and local $\delta^{18}O_{sw}$ of Rohling et al. (2021) and
Gaskell et al. (2023) respectively. The calculation was performed twice, both with and without a
pH/$[CO_3^{2-}]$ effect on $\delta^{18}O$ (the former using the mean planktonic foraminiferal slope of Gaskell et
al. (2023) and the $[CO_3^{2-}]$ record of Zeebe & Tyrrell (2019)).
When evaluating the paleotemperature reconstructions, we define whether or not the two proxy
systems agree within uncertainty by determining if the root sum of squares of the two uncertainties
is smaller than the temperature difference between the two proxies. We then proceed to identify
possible drivers for the data deviating from the expected Mg/Ca and $\delta^{18}O$ relationship by
evaluating the age of the sample, regional changes in $\delta^{18}O$ seawater, depth ecology, and possible
species-specific offsets.



We note that all of the above corrections assume surface ocean conditions, while the dataset
contains a number of species that calcify at depth (Boscolo-Galazzo et al., 2021). Given the
uncertainties surrounding past changes in vertical pH and $\delta^{18}O_{sw}$ profiles, we do not attempt to
account for this in our data analysis but note that this consideration should be borne in mind when
interpreting data from deep-dwelling species.

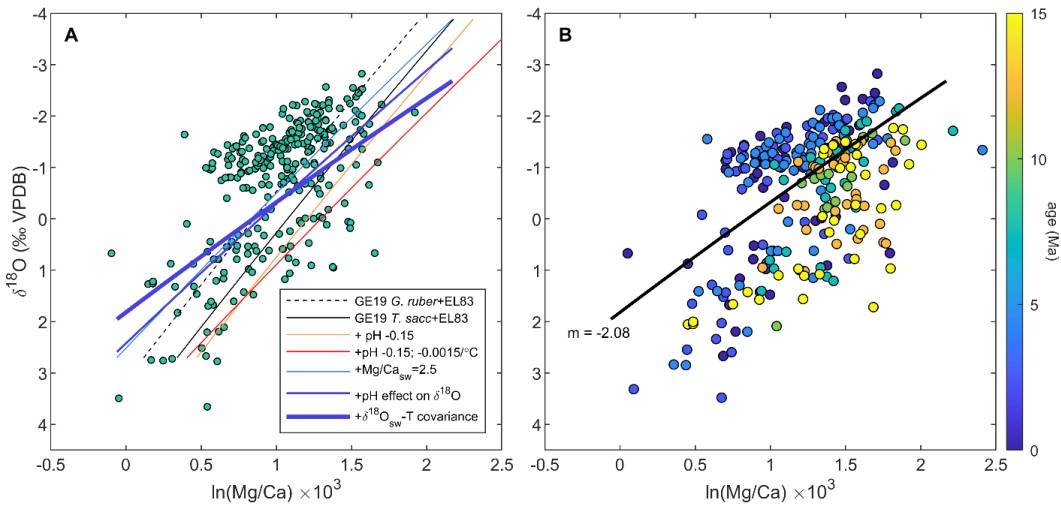


**Fig. 2**. Raw $\delta^{18}O$ plotted against Mg/Ca for all samples presented here. (A) Several possible expected
Mg/Ca-$\delta^{18}O$ slopes are shown for comparison, including that for *G. ruber* and *T. sacculifer* (at constant pH)
in the modern ocean (solid and dashed black lines respectively). The additive impact of other nonthermal
controls are then explored using the *G. ruber* calibration as an example, specifically, the impact of: a whole-
ocean pH shift of 0.15 units (orange line), accounting for the covariation of pH and temperature (driven by
the temperature-dependent dissociation of water, red line), seawater Mg/Ca half of the present day value
(thin blue line), the theoretical impact of pH on $\delta^{18}O$ (blue line), and the covariance of temperature and
$\delta^{18}O_{sw}$ in the modern ocean (thick blue line). The length of each line depicts the expected Mg/Ca and $\delta^{18}O$
change across the same temperature range in each case (5-35°C). All calculations assume $\delta^{18}O_{sw} = 0$‰. (B)
As in panel A, except with the long-term whole ocean changes in pH, Mg/Ca$_{sw}$, and $\delta^{18}O_{sw}$ subtracted out
of the raw proxy values (see text, using the multispecies calibration of Gray & Evans (2019) in the case of
the Mg/Ca corrections), i.e., accounting for the impact of these non-thermal Mg/Ca and $\delta^{18}O$ controls.
Sample age is shown as a function of colour.





## 3. Results


Our basic expectation is that higher Mg/Ca should relate to more negative $\delta^{18}O$ values for warmer
temperatures, and *vice versa* for colder temperatures. Despite the large number of variables
included, the dataset as a whole shows a significant correlation (Fig. 2A; $R^2 = 0.37$, RMSE = 1.01,
p <<0.01) between $\delta^{18}O$ and ln(Mg/Ca). Hence, the $\delta^{18}O$-Mg/Ca covariance can be considered a
robust feature over the past 15 Myr for the majority of the species analyzed and across the study
sites. Nonetheless, there is a high degree of scatter in the data which suggests that the temperature
signal which should lead Mg/Ca and $\delta^{18}O$ data to change consistently in opposite directions is
affected by other factors. Our exercise of generating theoretical Mg/Ca-$\delta^{18}O$ relationships (Fig.
2A), exploring how the relationship between the two proxies might change through space and time,
provides a qualitative indication as to whether the scatter can be attributed to long term non-
thermal factors generally corrected for when using the $\delta^{18}O$ and Mg/Ca proxies. The substantial
differences between these expected relationships suggests that this is likely to be the case (Fig.
2A). In particular, Fig. 2A suggests that both a pH effect and Mg/Ca$_{sw}$ changes through time may
explain a substantial degree of the variability observed in the dataset compared to the modern
relationships (compare the coloured and black lines). Nonetheless, accounting for these long-term
biases alone in the raw dataset does not remove the scatter (Fig. 2B), suggesting the importance of
additional factors, such as vital effects and regional variations in $\delta^{18}O_{sw}$. Therefore, we next
convert the raw proxy values into temperature including a correction for regional variations in
$\delta^{18}O_{sw}$. Data converted into temperature, along with 95% confidence intervals, are shown in
Figure 3. In this plot 62% of the data points fall within uncertainty, confirming that a high degree
of variability in the raw data can be effectively explained and accounted for by correcting for the
known spatially and temporally varying non-thermal effects influencing both proxies.






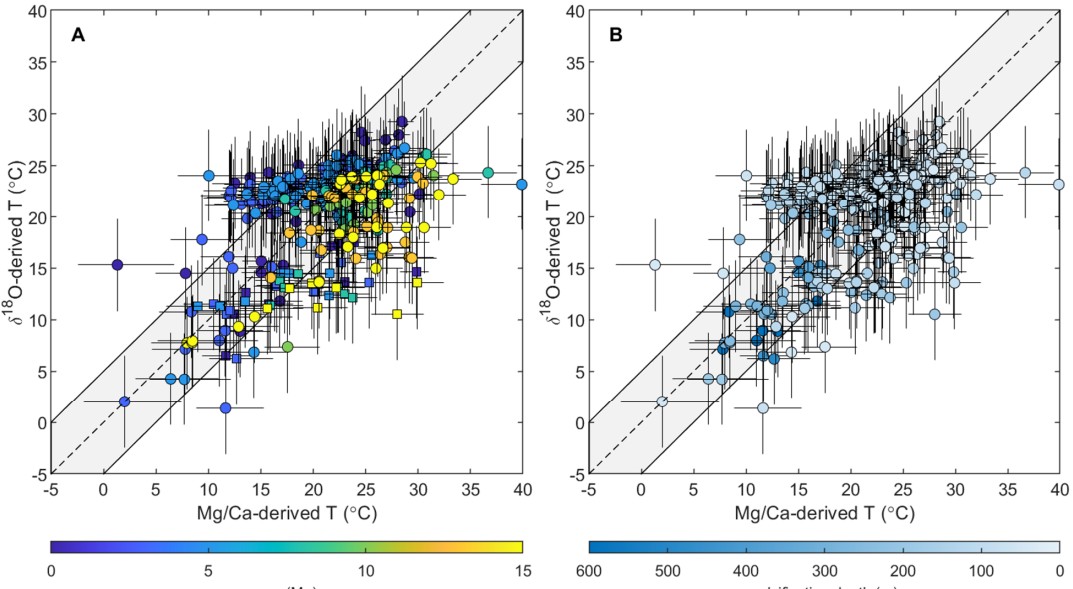


**Fig. 3**. $\delta^{18}O$ versus Mg/Ca-derived paleotemperatures plotted as function of age (A) and calcification depth (B), accounting for the impact of whole ocean and regional changes in $\delta^{18}O_{sw}$ following Gaskell *et al.* [2022] and the bayfox $\delta^{18}O$-temperature calibration [Malevich *et al.*, 2019], whole ocean changes in Mg/Ca$_{sw}$ and pH on Mg/Ca using *MgCaRB* [Gray & Evans, 2019], and including a pH correction on $\delta^{18}O$ using the mean planktonic foraminifera slope [Gaskell *et al.*, 2023]. Fully propagated uncertainties in both proxies are shown, incorporating analysis, calibration, pH, Mg/Ca$_{sw}$, $\delta^{18}O_{sw}$/salinity (see text for details). Site 516 data are shown with square symbols in panel A.


We find that deep-dwelling and surface-dwelling species fall within uncertainty in terms of the
broad degree of agreement between Mg/Ca and $\delta^{18}O$ derived temperatures, hence different depth-
habitat ecologies or calcification environments do not represent a systematic source of offset *per*
*se* (Fig. 3B). Indeed, all the species displaying a temperature offset between the two proxies are
surface-dwellers (0-200 m depth, Boscolo-Galazzo et al., 2022), with deep-dwellers characterized
by inter-proxy agreement in almost all cases (Fig. 3B). Additional explanations are required for
those temperature reconstructions that, all non-thermal and spatial controls included, still plot
outside their propagated error uncertainty. The data points outside the combined proxy



uncertainties either display >5° C warmer (colder) $\delta^{18}$O (Mg/Ca) temperatures (upper left area of
the plot) or >5°C warmer (colder) Mg/Ca ($\delta^{18}$O) temperatures (bottom right area of the plot) (Fig.
3). Figure 3A clearly shows that the large majority of the outliers in the upper left part of the plot
(warmer $\delta^{18}$O temperatures or cooler Mg/Ca temperatures) are species of late Miocene to modern
age, while the outliers in the bottom right part of the plot (warmer Mg/Ca temperatures) are mostly
older species of middle Miocene age or from mid-latitude Site 516 (squares in Fig. 3C). We suggest
and discuss possible main sources for these offsets between Mg/Ca and $\delta^{18}$O temperatures below.
**4. Discussion**
4.1 Diagenesis
Diagenesis is known to alter the test chemistry of foraminifera in three main ways: partial
dissolution, overgrowth, and recrystallization (Edgar et al., 2015). The trace element and isotopic
composition of tests react differently to these diagenetic processes. The trace element composition
of foraminiferal calcite may be susceptible to partial dissolution because it is inhomogeneous
(Fehrenbacher et al., 2014), which decreases trace element ratios in species with high and low-Mg
regions (Dekens et al., 2002; Edgar et al., 2015; Rongstad et al., 2017), resulting in lower
temperature reconstructions. Overgrowth and recrystallization have been shown to add both low-
Mg and high-Mg diagenetic calcite, potentially impacting the original signal in opposite directions
(Branson et al., 2015), although Mg/Ca is relatively robust to this type of diagenesis, at least in
certain circumstances (Staudigel et al., 2022). The oxygen isotopic composition of planktonic
foraminiferal tests is well known to be very sensitive to overgrowth and recrystallization (e.g.
Sexton et al., 2006), whereby the addiction of diagenetic calcite, or the replacement of the original
calcite with diagenetic calcite precipitated at the seafloor, can significantly alter the original
isotopic signal shifting it to more positive values (Pearson, 2012; Edgar et al., 2015).





The Boscolo-Galazzo, Crichton et al. (2021) dataset spans 15 million years and includes sites with
different average preservation of foraminiferal tests and oceanographic settings. When the data are
regressed against each other (Fig. 2B), we find a total of 26 data points characterized by oxygen
isotope values more positive than expected from their Mg/Ca values (Fig. 2B; Supplementary
Table 1), resulting in $\delta^{18}$O temperatures >5° C colder than Mg/Ca temperatures (outside the error
envelope) (Fig. 3; Supplementary Table 1). Twenty-one of these data points were from the older
time slices (12.5 and 15 Ma), one from the 7.5 Ma time slice and four from the core-top of Site
U1338 (Supplementary Table 1).
As the majority of these datapoints were characterized by Mg/Ca values of ~1.5-2 ln(Mg/Ca) (Fig.
2B), this yielding more reasonable Mg/Ca than $\delta^{18}$O temperatures for these sites/time intervals
(Supplementary Table 1),  the observed offset is most likely best attributed to diagenetic
overgrowth/recrystallization, shifting oxygen isotopes towards more positive values without
affecting Mg/Ca to the same extent. A recent study compared typical Mg/Ca-$\delta^{18}$O from
recrystallized planktonic foraminifera with chemical diffusive models simulating early diagenetic
processes in calcite (Staudigel et al., 2022). According to that study, in a closed system, the bulk
$\delta^{18}$O value will be altered faster than the Mg/Ca, regardless of what partitioning coefficient is used
for Mg, leading to a progressive shift to more positive $\delta^{18}$O values leaving Mg/Ca virtually
unchanged (Staudigel et al., 2022).
The datapoints presenting $\delta^{18}$O overprinted by overgrowth/recrystallization were distributed
across most of the study sites (except for Sites 871/872), but they were more common at Site
U1490 and U1489 (15/26) which are characterized by inferior preservation compared to the others
(Boscolo-Galazzo, Crichton et al., 2021). The core-top samples at Site U1338 show clear signs of
dissolution with highly fragmented tests. These datapoints presented the lowest Mg/Ca values in



the dataset (1.68 and 0.91 mmol/mol) with temperatures from Mg/Ca lower than from $\delta^{18}$O. This
suggests that partial dissolution and recrystallization affected both Mg/Ca and $\delta^{18}$O in this sample,
but Mg/Ca more so.
Overall, our scrutiny for diagenesis of the Boscolo-Galazzo, Crichton et al., (2021) dataset is
consistent with previous studies suggesting that $\delta^{18}$O values are more easily affected by
recrystallization than Mg/Ca (Sexton et al., 2006; Staudigel et al., 2022; John et al., 2023), similar
to other trace element systems (Edgar et al., 2015).
Based on the considerations above we excluded the affected 26 datapoints from the subsequent
analysis as being characterized by a stronger diagenetic overprint than the rest of the dataset
(Supplementary Table 1). Removing the affected datapoints in some but not all the cases equated
to removing a whole sample (Supplementary Table 1). This is because of variable diagenetically
offset $\delta^{18}$O values from different species in a sample, as  observed elsewhere (Sexton et al., 2006;
Edgar et al., 2015). The approach used here to reconstruct $\delta^{18}$O temperature shows that for the
majority of the study dataset a diagenetic offset would be comprised within the propagated error
envelope (~2-2.5°C) (Fig. 3) and comparable or not distinguishable from an offset deriving from
poorly constrained $\delta^{18}O_{sw}$.
4.2 Regional scale spatial heterogeneity in seawater chemistry
Once converted into temperatures the dataset shows an overall good agreement of Mg/Ca-$\delta^{18}$O
data, consistent with temperature being a dominant controller of both proxies through time and
across the broad geographical area investigated (Fig. 3). This suggests that, by and large, the
seawater corrections applied for the local and global changes in ocean chemistry are adequate,
although for one site this may not hold true. Site 516 is a mid-latitude site (south-west Atlantic,





30°S) characterized by a modern sea surface temperature around 20°C (Fig. 1), most of the data
points from this site have very positive $\delta^{18}O$ values associated with high Mg/Ca (Fig. 2B;
Supplementary Table 1). Once converted, this results in $\delta^{18}O$ temperatures that are too cold (12-
17°C) compared to both modern (given long-term warming since the Miocene is not expected at
any of these sites) and the equivalent Mg/Ca temperatures from the same samples, which are
around 21-25°C (Fig. 3A; Supplementary Table 1). We do not attribute this mismatch to diagenesis
for a number of reasons. First, Site 516 is characterized by a very good test preservation, much
better than at Site U1489 and U1490 for which diagenesis extensively affects the middle Miocene
samples; second, this mismatch is observed in the entire dataset through samples spanning the
middle Miocene to modern; third, the mismatch is observed for surface-dwelling species only,
with deep-dwellers characterised by $\delta^{18}O$ - Mg/Ca in good agreement. Because of its location, Site
is situated in an area of complex surface hydrography, as it sits at the confluence between the
warm Brazil Western Boundary Current and the cold Falkland (Malvinas) Current spinning off
from the Antarctic Circumpolar Current (e.g., Jonkers et al., 2021). Compared to the subtropical
gyres, where many of the study samples come from, a large degree of spatial variability of surface
water physical-chemical properties can be expected on a seasonal and multiannual scale. As such,
we suggest that the mismatch between $\delta^{18}O$ and Mg/Ca observed in surface dwelling species at
Site 516 may result from changeable surface water properties from the mixing of two very different
water masses creating deviations in pH, salinity and $\delta^{18}O_{sw}$ beyond those that are typical in
stratified open ocean environments and that are difficult to correct for. Sites with a changeable
hydrography such Site 516, may hence not be ideal for the application of geochemical proxies
affected by seawater chemistry, unless changes in seawater chemistry at the site can be
reconstructed directly.



### 4.3 Species-specific offsets

The third and largest source of mismatch that we consider is the occurrence of species-specific offsets, particularly in Mg/Ca given that, in general, the relative degree of inter-species Mg/Ca variability is greater than for shell oxygen isotope composition (e.g., compare Pearson, 2012; Gray & Evans, 2019, Regenberg et al., 2009). Here, by the phrase "species-specific offset" we refer to atypical geochemical signatures which characterize certain species likely as a result of processes linked to the organism's metabolism or calcification (i.e., vital effects). When regressing the data against each other, we find several species presenting systematically offset Mg/Ca values (e.g., Fig. 2B; Supplementary Table 1). The occurrence of such offset Mg/Ca values is not evenly distributed across species, but is shared among related species in both spinose and non-spinose groups (Figs. 4-5; Supplementary Table 1). Specifically, the spinose offset species *Orbulina universa* (2/5 specimens), *O. suturalis* (3/6) and *Praeorbulina glomerosa* (1/1) present high Mg/Ca ratios compared to their $\delta^{18}O$ values and Mg/Ca of other species (Fig. 6E). We also find that *Globigerinella siphonifera* (6/7), *G. calida* (1/1), *G. praesiphonifera* (3/4) and *Globigerina bulloides* (5/6) have offset Mg/Ca-$\delta^{18}O$ values, largely being characterized by higher than expected Mg/Ca, although three *G. siphonifera* data points show lower Mg/Ca (Figs. 4, 6K; Supplementary Table 1). Among non-spinose species, offset species are: *Neogloboquadrina humerosa* (11/12), *N. acostaensis* (3/3), *Pulleniatina obliquiloculata* (5/6), *P. praecursor* (3/5), *P. primalis* (4/6), *Sphaeroidinella dehiscens* (4/4), and *Sphaerodinellopsis paenedehiscens* (5/10), which have Mg/Ca values lower than expected for their oxygen isotope composition (Figs. 5, 6 G, I; Supplementary Table 1).




These results highlight for the first time the occurrence of similarly offset Mg/Ca values for
ancestor-descent species belonging to the same lineage as well as, in the case of *Pulleniatina,* to a
whole lineage descending from *Neogloboquadrina* (Figs. 4-6).

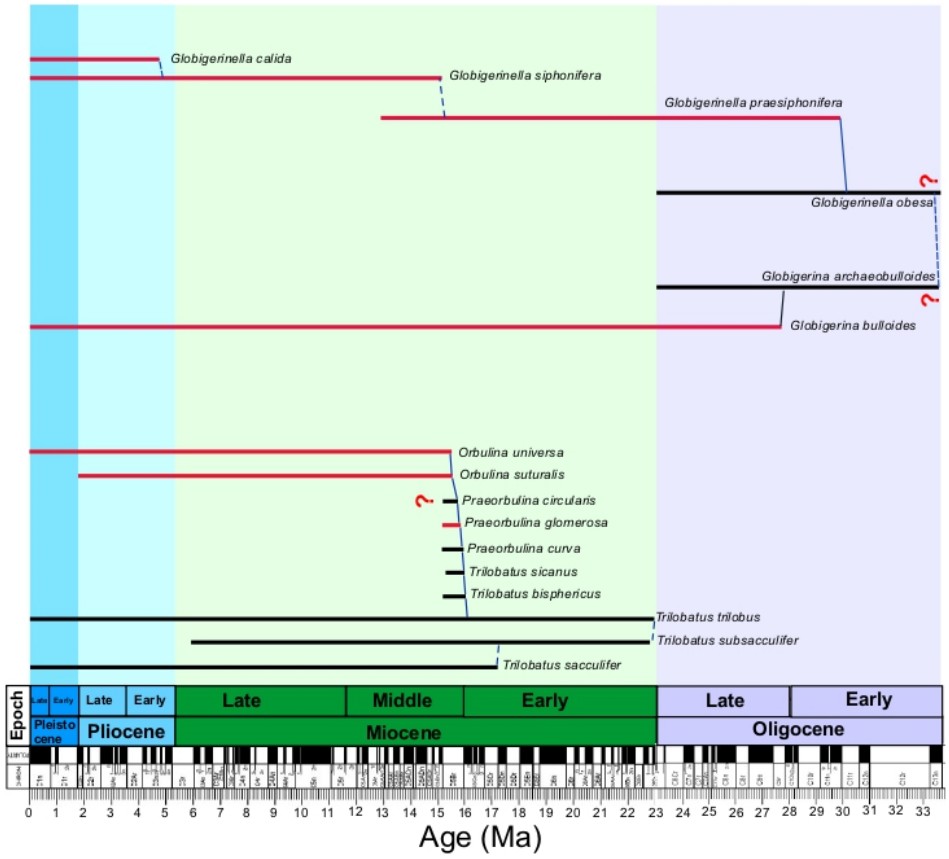


**Fig. 4.** Phylogenetic relationships of offset spinose-species. Shown here are the species discussed
in the text, their most closely related species and the ancestors. Red lines indicate species offset in
Mg/Ca in the study dataset relative to a multispecies calibration approach, black lines indicate non-
offset species. Red question marks indicate the lack of Mg/Ca data for a given species in the dataset
presented here. Phylogeny after Aze et al. (2011) and Spezzaferri et al. (2018). The phylogenetic
chart was generated using Mikrotax (Huber et al., 2016; www.mikrotax.org/pforams). The
reference time scale in the figure is the Astronomical time scale of Lourens et al. (2004), until base
of Chron C6Cn.2n, and Pälike et al. (2006), from top Chron C6Cn.3n until base C13n.





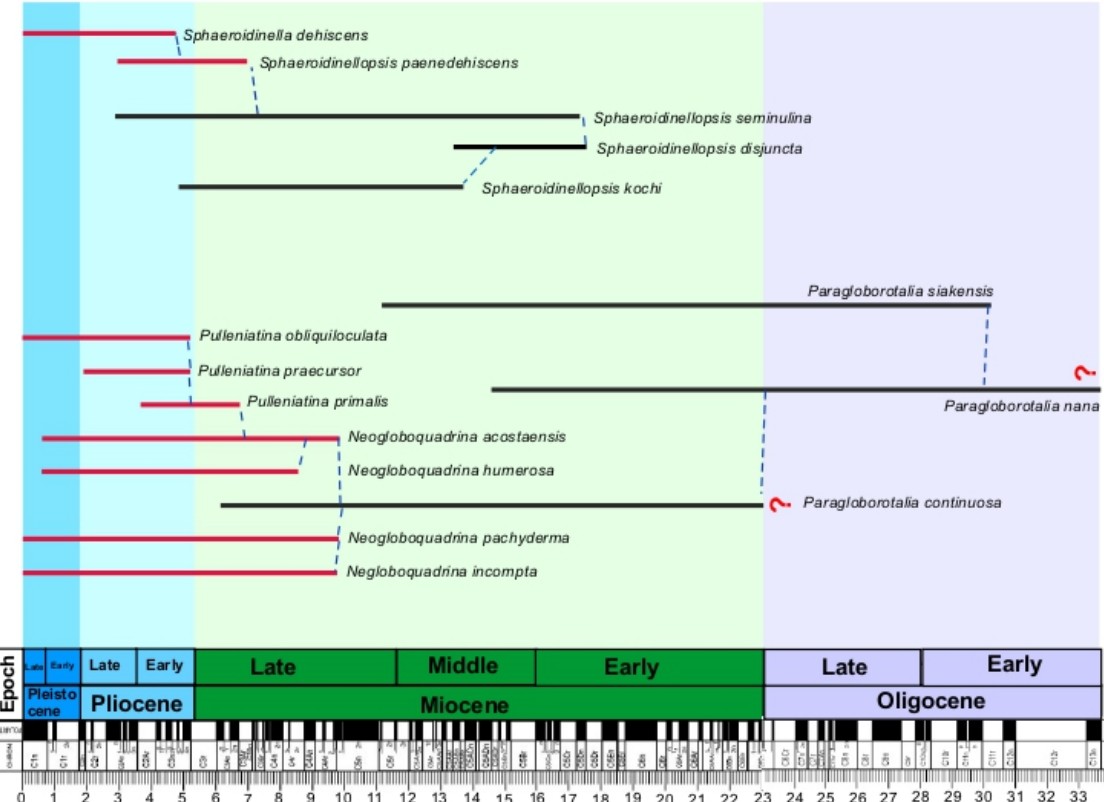

456

**Fig. 5.** Phylogenetic relationships of offset non-spinose species. Shown here are the species discussed in the text, their most closely related species and the ancestors. Red lines indicate species offset in Mg/Ca in the study dataset relative to a multispecies calibration approach, black lines indicate non-offset species. Red question marks indicate the lack of Mg/Ca data for a given species in the study dataset. Phylogeny after Aze et al. (2011), Leckie et al. (2018) and Fabbrini et al. (2021). Phylogeny chart generated using Mikrotax (Huber et al., 2016; www.mikrotax.org/pforams). The reference time scale in the figure is the Astronomical time scale of Lourens et al. (2004) until base of Chron C6Cn.2n and Pälike et al. (2006), from top Chron C6Cn.3n until base C13n.


Divergent Mg/Ca values for *G. siphonifera* and *O. universa* have previously been reported
(Opdyke and Pearson, 1995; Anand et al., 2003; Friedrich et al., 2012). In the case of *O. universa*,
the offset may be related to pH change in the foraminiferal microenvironment due to symbiont
photosynthetic activity (Eggins et al., 2004) or changes in seawater pH, with Mg/Ca of the test



increasing by as much as 6±3% for each 0.1 unit decrease in pH (Lea et al., 1999; Russell et al.,
2004). pH-related vital effects are reported for other spinose species of planktonic foraminifera
such as *Globigerina bulloides* (Lea et al., 1999; Davis et al., 2017), which is related to the genus
*Globigerinella* (Fig. 4).
Among the neogloboquadrinids, *N. acostaensis* and its descendent *N. humerosa* have the most
clearly expressed offset with low Mg/Ca values. In contrast, *Neogloboquadrina pachyderma* and
*N. incompta* are not offset in the study dataset, perhaps simply because of the limited amount of
data (one data point each). More broadly, a Mg/Ca offset compared to other species has been
reported in the literature (Davis et al., 2017). *Neogloboquadrina dutertrei*, *N. incompta*, *N.*
*pachyderma* and *Pulleniatina obliquiloculata* have been shown to be characterized by much lower
trace element concentrations (Mg-Ba-Zn/Ca) in the adult portions of their shells (crust and cortex),
so that a greater amount of adult versus early ontogenetic calcite leads to low trace element values
in bulk shell analysis (Jonkers et al., 2012; Davis et al., 2017; Fritz-Endres & Fehrenbacher, 2021).
The low Mg/Ca of crust and cortex have been found to be independent of ambient temperature in
cultured *Neogloboquadrina* (Davis et al., 2017) and are found in specimens collected both in
surface waters and at depth (Jonkers et al., 2021), indicating that the low Mg/Ca is not acquired
due to calcification in deeper, colder waters of the crust/cortex portion of the shell, although a
greater incidence of crusts is reported for colder waters (Jonkers et al., 2021). In our dataset,
*Neogloboquadrina*, *Pulleniatina* and *Sphaeroidinella/Sphaerodinellopsis* are all characterized by
a thick crust or cortex suggesting their Mg/Ca are biased by low Mg adult calcite being
quantitatively predominant, which is further corroborated by their Mg/Ca being unrelated to
temperature in our data and consistently falling outside of the $\delta^{18}$O-derived temperatures even
accounting for the combined uncertainty of the two proxies (Fig. 6 G-H, I-J).



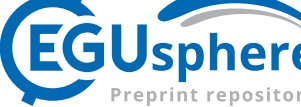

The majority of the data points from the offset spinose and non-spinose species results in
temperature differences between the two proxies greater than 5°C when using the multi-species
calibration from Gray and Evans (2019) as described in §2.4.2, hence outside the calculated error
envelope taking all the non-thermal factors discussed above into account (Fig. 6). A similar
temperature offset is not apparent for other lineages such *Trilobatus trilobus – Trilobatus*
*sacculifer* and *Globigerinoides subquadratus – G. ruber*, to which the same treatment to the offset
spinose and non-spinose species was applied (§2.4.2, Figs. 6 A-D). Hence, we attribute the offset
temperatures to the atypical Mg/Ca signatures described above in the affected species, in turn
resulting from biology/ecology dependent vital effects shared within a lineage and between related
lineages (Figs. 4-5).
When a species-specific calibration for *Neogloboquadrina pachyderma* is applied to descendent
species/lineages and sister taxa (Fig. 5) the offset is successfully corrected for all the
*Neogloboquadrina* and *Pulleniatina* species which effectively no longer produce offset
temperatures (Figs. 6, 7). *Vice versa*, we only observe a minor improvement when applying the
*Orbulina universa* calibration to the *Prearobulina-Orbulina* lineage, with most temperature data
points remaining offset (Figs. 6, 7), albeit in the opposite direction. This may imply that the *O.*
*universa* laboratory calibrations require revision for application to fossil samples. No large
difference is observed when applying the *Trilobatus sacculifer* calibration to ancestor-descendent
species in the genus *Trilobatus* (Figs. 5-7) although we recommend doing so, given that no Mg/Ca-
pH effect is known for this genus, in contrast to (e.g.) *G. ruber*.
Overall, this exercise demonstrates that the majority of the data points characterized by proxy-
proxy disagreement (Fig. 3) are from the lineages: *Praeorbulina-Orbulina*, *Globigerina-*
*Globigerinella, Neogloquadrina, Pulleniatina, Sphaeroidinellopsis-Sphaeroidinella* (Fig. 6). We

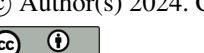


find that using a "nearest descendant" approach in the choice of temperature calibration improves

the agreement between $\delta^{18}$O and Mg/Ca temperatures for the neogloboquadrinids and

pulleniatinids (Fig. 7). At the same time, it enables us to identify "problematic" species and

lineages which require further investigations before being used for temperature reconstructions

(Fig. 7).

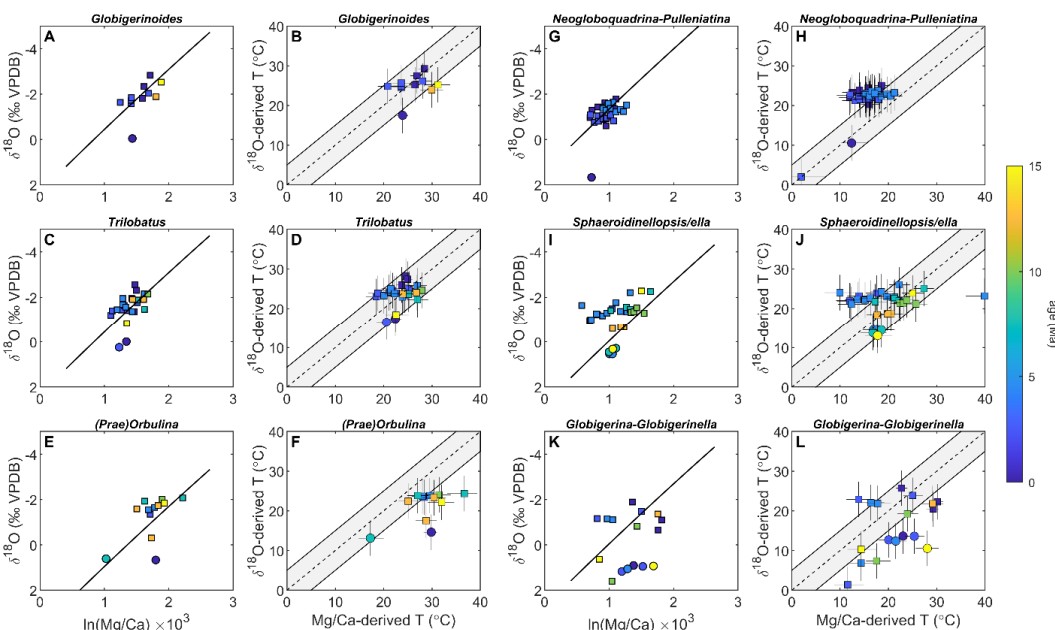

**Fig. 6** $\delta^{18}$O versus Mg/Ca and proxy-derived paleotemperature estimates for the ancestor-descendent species *Globigerinoides subquadratus – G. ruber* (panels A and B), *Trilobatus trilobus – T. sacculifer* (panels C and D), *Praeorbulina glomerosa – Orbulina suturalis – O. universa* (panels E and F), *Neogloboquadrina acostaensis – N. humerosa, N. pachyderma – N. incompta, N. acostaensis – Pulleniatina primalis – P. praecursor – P. finalis* (panels G and H), *Sphaerodinellopsis seminulina – S. paenedehiscens – Sphaeroidinella dehiscens, S. seminulina – S. kochi* (panels I and J), *Globigerinella praeshiphonifera – G. siphonifera – G. calida* and *Globigerina bulloides* (panels K and L). Circle-symbols indicate data points from Site 516. Raw proxy values are given with the long-term non-thermal controls on Mg/Ca subtracted out (as in Fig. 2), as well as an estimate of paleotemperature (as in Fig. 3). The black lines depict one possible estimate of the expected slope between $\delta^{18}$O and Mg/Ca (the blue line from Fig. 2), adjusted to approximately match the location of the data by shifting them in the direction of $\delta^{18}$O. Datapoints which are considered strongly affected by diagenesis are not included in this plot. Note that one datapoint in panel G falls outside of the plot area.

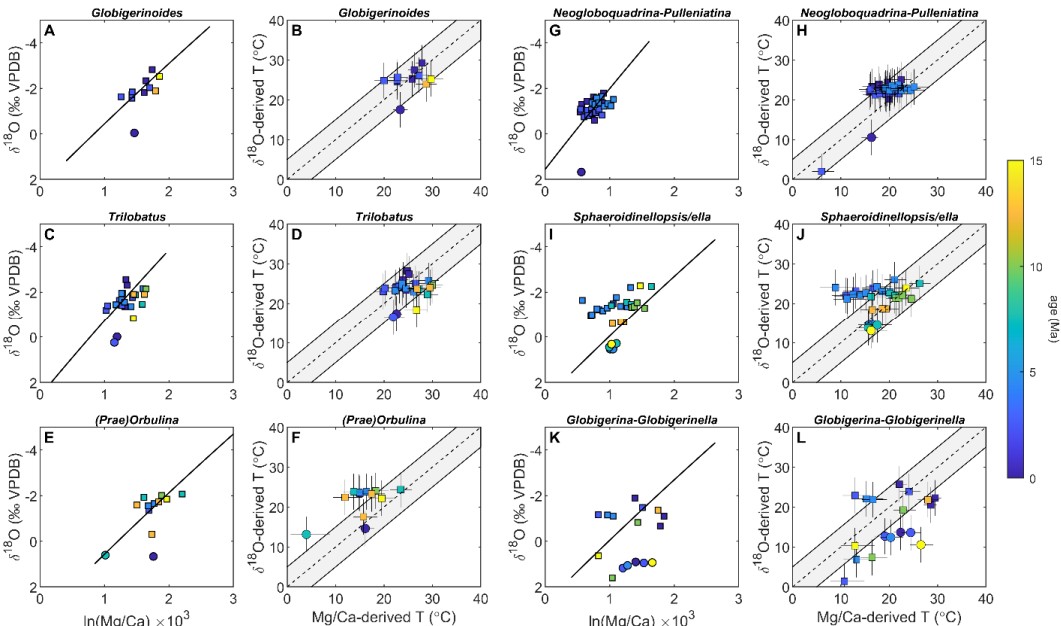

**Fig. 7**. As in Figure 4, except extending the use of species-specific calibrations to all species in a lineage in the case of the *Trilobatus trilobus – T. sacculifer* (panels C and D), *Praeorbulina glomerosa – Orbulina suturalis – O. universa* (panels E and F), *Neogloboquadrina acostaensis – N. humerosa, N. pachyderma – N. incompta, N. acostaensis* and between related lineages in the case of N*eogloboquadrina* and the *Pulleniatina primalis – P. praecursor – P. finalis* lineage (panels G and H).

Once all the potential sources of offset described above are taken into account, the species-specific calibration for *Neogloboquadrina* is applied, and the data points which are still offset removed from the dataset, the agreement between the two proxies increased from 62 to 91% of data points falling within the combined uncertainties of the proxies (Fig. 8).





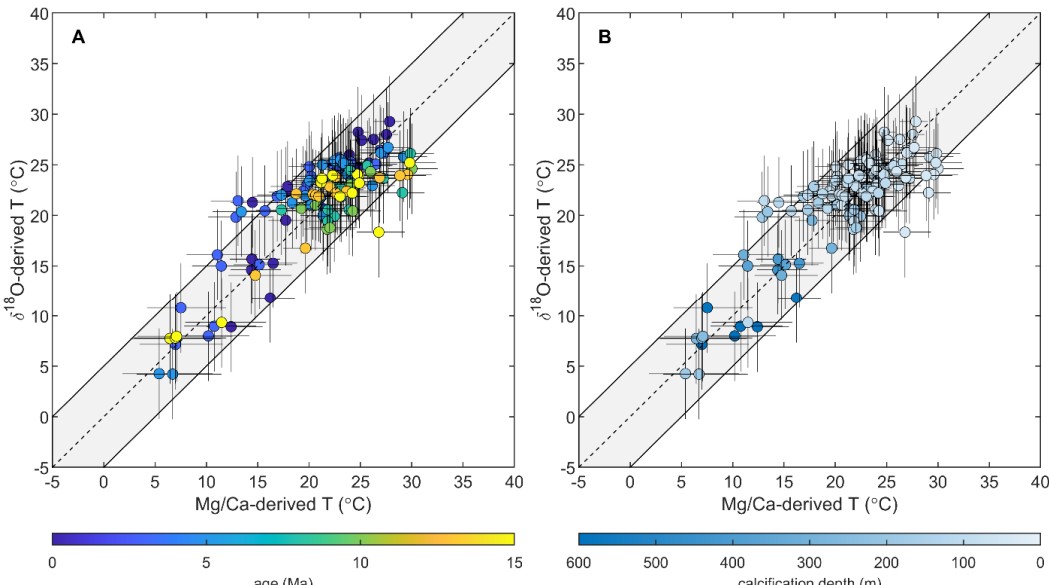

**Fig. 8.** As Fig. 3, with all offset lineages (§ 4.3; specifically, those that remain offset following the application of lineage-specific calibrations where possible) and diagenetically compromised (§ 3.1) samples removed. Removing these samples leaves 170 data points, of which 91% fall within the combined uncertainty of Mg/Ca-$\delta^{18}$O agreement.

## 4.4. Planktonic foraminiferal Mg/Ca offsets as an expression of evolution

The analysis of the Boscolo-Galazzo, Crichton et al. (2021) dataset performed here, allows offsets

through ancestor-descendent species to be tracked for the first time, and the time of their

appearance to be assessed. In this way, an attempt can be made to interpret offsets as the

geochemical expression of evolutionary new biochemical pathways or ecological strategies in

emerging species.

For spinose species, the observed high Mg/Ca offset is shared by ancestor-descendent species such

as *Globigerinella praesiphonifera - G. siphonifera* and *Praeorbulina glomerosa - Orbulina*

*suturalis - O. universa* (Fig. 4). *Globigerina bulloides* shares the same type of offset with



*Globigerinella* and a common ancestor (*Globigerina archaeobulloides*) in the earliest Oligocene
(~33.5 Ma) (Spezzaferri et al., 2018) (Fig. 4), suggesting that for this group the offset may go back
to at least the early globigerinids of the Paleogene. The genus *Preaorbulina* originated from the
genus *Trilobatus* at about 16 Ma (Fig. 4) (Pearson et al., 1997; Aze et al., 2011). *Trilobatus trilobus*
is the last common ancestor between the *Trilobatus* and *Prearobulina-Orbulina* lineages (Fig. 4),
and does not present offset Mg/Ca-$\delta^{18}$O values, similar to its modern descendants (Figs. 4, 6). This
suggests that the offset in *Praeorbulina-Orbulina* originated within the lineage and the
morphological changes associated with it, and carried on to the modern representative *O. universa*.
Spinose *Globigerinoides ruber* is also reported to be sensitive to pH changes (Kisakürek et al.,
2008; Evans et al., 2016b), *G. ruber* is not offset in the analyzed dataset (both in the raw data and
calculated temperatures), with ancestor-descendent *G. subquadratus-G. ruber* behaving similarly
to the *Trilobatus trilobus – T. sacculifer* lineage through time (Figs 6-7), suggesting that this non-
thermal effect is adequately accounted for in this case (Fig. 7) (i.e., the laboratory calibration are
applicable downcore into deep-time in correcting for this). The on-average higher Mg/Ca
displayed by offset spinose *Preaorbulina-Orbulina*, *Globigerinella* species and *G. bulloides* in the
study dataset may suggest a lower pH environment which we cannot directly account for (Fig.
2A). For *Globigerina* and *Globigerinella*, where a larger degree of scatter is observed (Fig. 6-7 K-
L), the offset maybe linked to an opportunistic behavior and capability to adapt to a broad range
of environmental conditions with variable pH (Weiner et al., 2015). This may in turn be related to
the complexity of genotypes association in both *G. bulloides* and *Globigerinella* species, with
different genotypes having different ecologies but almost identical morphologies (e.g., Weiner et
al., 2015; Morard et al., 2024).





We find that the low Mg/Ca offset is shared between ancestor-descendent lineages
*Neogloboquadrina-Pulleniatina* but possibly not between the lineages *Paragloborotalia-*
*Neogloboquadrina* and is not shared between *Sphareoidinellopsis-Sphaerodinella* (Fig. 5).
*Neogloboquadrina* evolved from *Paragloborotalia continuosa*, in the late Miocene, at about 10
Ma (Fig. 5). While paired Mg/Ca – $\delta^{18}$O measurements for *P. continuosa* are not available, paired
Mg/Ca – $\delta^{18}$O measurements for *Paraglobortalia siakensis,* a species older than *P. continuosa,* do
not show a low Mg/Ca offset (Supplementary Table 1). This may indicate either that the occurrence
of low Mg/Ca crust/cortex started with *P. continuosa*, the youngest representative of the genus
*Paragloborotalia* in our study, or with the neogloboquadrinids. The genus *Pulleniatina* evolved
from *Neogloboquadrina acostaensis* at about 6.5 Ma (Pearson et al., 2023) (Fig. 5), by modifying
the chambers arrangement and progressively developing a cortex. *Pulleniatina* may have inherited
the capability to thicken its test from *Neogloboquadrina* and modified it into a cortex (Pearson et
al., 2023). The occurrence of a low Mg/Ca cortex in *Pulleniatina* appears to start from the most
ancient representative of this group, *P. primalis* (Figs. 5-6) and continues to the modern. Similar
to *Paragloborotalia*, middle Miocene *Sphaeroidinellopsis kochi* and *S. seminulina* are not
characterized by an offset to low Mg/Ca values. Nonetheless, an offset is observed in 5/10
specimens for late Miocene – early Pliocene S. *paenedehiscens*, and always occurs for its
descendent *Sphaeroidinella dehiscens* (Fig. 7). Our analysis shows how the occurrence of a low
Mg/Ca offset in planktonic foraminifera becomes progressively rarer going back in time, in
parallel with the rarity of crust/cortex features. The occurrence of a crust/cortex is commonly
observed in non-spinose modern planktonic foraminifera, however, only two early to middle
Miocene genera are known to produce crusts (*Globoconella* and *Paragloborotalia*) and only one
is known to produce cortex (*Sphaerodinellopsis*).



It is tempting to put the pattern of emergence of Mg/Ca offsets in relationship with changes in
ocean chemistry and global climate over the last 15 Myr. In particular, the offset spinose species
are mostly tropical and evolved during a time when mean ocean pH was more than 0.1 pH unit
lower than preindustrial (Rae et al., 2021). Further, the *Preorbulina - Orbulina* plexus evolved at
about 16 Ma, in coincidence with a drop in ocean pH likely linked to the global warmth of the
Miocene Climatic Optimum (Rae et al., 2021). The particularly high Mg/Ca signature of this group
of species, along with their evolutionary timing, may testify their ability to withstand tropical
surface waters more undersaturated than today thanks to changes in the biomineralisation pathway
as a consequence of their evolution during the Miocene Climatic Optimum.
The evolution of the offset non-spinose species happened several millions of years later during the
long-term cooling trend of the last 10 Myr. The offset species occur across tropical to high latitude
areas and mixed-layer (*Sphaeroidinellopsis, Sphaeroidinella*) to intermediate (*Neogloboquadrina,*
*Pulleniatina*) depth habitats. The ability to develop crust/cortex in species evolving over the last
10 Myr might have been of advantage as the global ocean was becoming progressively colder and
denser, in a similar way to the observed increases in shell-mass across Pleistocene glacial cycles
(e.g., Zarcogiannis et al., 2019).
The last 10 Myr were also characterized by decreasing concentration in $Ca_{sw}$ ($[Ca^{2+}]$) in step with
global cooling, reaching concentrations half those of the middle Miocene in the modern (Zhou et
al., 2021). As a consequence, $Mg/Ca_{sw}$ doubled over the past 5 Ma (Evans et al., 2016b). With
decreasing $[Ca^{2+}]$ and increasing $[Mg^{2+}]$ in seawater over the Neogene (Brennan et al., 2013; Evans
et al., 2016b; Zhou et al., 2021), some species may have started to more actively control the Mg/Ca
ratio at their biomineralisation site, e.g., by proportionally decreasing the active transport of $Mg^{2+}$,
in order to buffer against the effects of the higher seawater Mg/Ca, and to keep the outer parts of





their shell with low Mg/Ca and thus more resistant to dissolution. Hence, the low Mg/Ca offset
observed in the modern and fossils non-spinose species above might be linked to their evolutionary
emergence during a time of changing ocean physical-chemical properties, which may have
promoted the evolution of thicker tests with a different elemental chemistry making them less
buoyant and resistant to dissolution.
**5. Conclusions**
We analyzed a multispecies planktonic foraminiferal $\delta^{18}O$ and Mg/Ca dataset spanning the last 15
million years at multiple locations to test whether temperature is the main controller of both proxies
and assess the major overprinting factors through time, space and for species with very distinct
ecologies. Once diagenesis and possible regional hydrographic factors are taken into account, we
find that species-specific offsets not accounted for in our calibration strategy remain a source of
mismatch between the two proxies. Specifically, *Globigerina*, *Globigerinella*, *Praeorbulina* and
*Orbulina* species are consistently offset, with Mg/Ca values on average higher than expected.
Conversely, non-spinose *Neogloboquadrina*, *Pulleniatina* and *Sphaeroidinellpsis-Sphaeroidinella*
appear consistently offset with low Mg/Ca. The appearance of these geochemical offsets is linked
to the origination of new clades, and is then shared between ancestor-descendent species, such that
we were able to track their evolutionary history. The variable offset in *Globigerinella* may go back
to the early globigerinids of the Paleogene and is probably related to the opportunistic behavior of
this group leading to a wider-range of habitat conditions. The high Mg/Ca offset in *Orbulina* starts
with *Praeorbulina* in the middle Miocene, while a low Mg/Ca offset is typical of groups evolving
in the late Neogene characterized by a crust or cortex. This pattern suggests that the offsets
observed in modern species may be a legacy of their parent groups originating millions of years
ago, when ocean properties were different from today.



Overall, our study highlights the power of the multispecies and multi-time slice dataset presented
here, enabling us to identify the evolutionary origin and timing of deviations in Mg/Ca-
temperature/pH relationships. Furthermore, our study demonstrates the robustness of Mg/Ca and
$\delta^{18}O$ proxies through geologic time when nonthermal factors (especially Mg/Ca$_{sw}$ and pH) are
accounted for. For example, virtually all *Globigerinoides* and *Trilobatus* Mg/Ca and $\delta^{18}O$-derived
temperature are within uncertainty of each other, highlighting the utility of these species for
paleoceanographic reconstruction. In addition, our analysis enables us to identify species/lineages
that should be treated with caution when interpreting Mg/Ca data, at the very least demonstrating
that care should be taken in selecting the calibration approach and highlighting the need for further
work in understanding the nonthermal controls on Mg incorporation into the shells of these
foraminifera.
**Code and Data availability**
All data are available as supplementary material of this paper. R and Matlab code to perform the
'MgCaRB' protocols are available on Github: https://github.com/willyrgray/ MgCaRB for R,
https://github.com/dbjevans/MgCaRB for Matlab.
**Author contributions**
E.M.M. performed trace element analysis; F.B.G conceptualized the paper; D.E. performed data
analysis; F.B.G and D.E. produced the figures; F.B.G and D.E. wrote the paper with contributions
from all authors.
**Competing interests**
The authors declare no competing interests.



**Acknowledgments**

This study was funded by Natural Environment Research Council (NERC) grant NE/N001621/1 to P.N.P. (F.B.G.); NERC grant NE/P016375/1 to participate in IODP Expedition 363 (P.N.P.); and NERC grant NE/N002598/1 to B.S.W. (E.M.M.). Marcin Latas assisted with sample preparation funded by an EU Marie Curie Career Integration Grant 293741 to B.S.W; F.B.G acknowledges support from Horizon 2020 Framework Programme (H2020-MSCA-IF-2020 101019438).

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
