# Peer review of "Exploring macroevolutionary links in multi-species planktonic foraminiferal Mg/Ca and"

_EGUsphere, 2024_

## Author Comment (AC1)

**Review rebuttal for:**

**Mg/Ca and δ$^{18}$O in multiple species of planktonic foraminifera from 15 Ma to Recent**

By Boscolo-Galazzo et al.

We thank the reviewers and editor for their comments and feedback on the manuscript. We agree with almost all of the points raised and have revised the text accordingly. Below we give a point-by-point response to the reviewers comments.

**Reviewer 1, Lennart de Nooijer**

*The statistical approaches should be explicitly described. This includes the exact way the regression analyses were done and the way the MC simulation was applied. Please provide the necessary details.*

To clarify the text we have rephrased the part which describes the Monte Carlo simulations specifically (lines 259-260). We give the details of uncertainty propagation and the sources of uncertainty that are propagated in Sec. 2.4.2.

*Line 254: do the 'calibration coefficients' refer to the uncertainty in the estimated constants from the regression analysis?*

We now clarify this on line 258.

*It would be interesting to know which of the sources of the uncertainty contribute most to the overall CI.*

The main sources of uncertainties are derived from: the calibration coefficients, Mg/Casw, pH, and in the case of d18O, the d18Osw. In the case of Mg/Ca, the magnitude of these uncertainties is approximately split between the uncertainty on the calibration coefficients and the uncertainty on the Mg/CaSW reconstruction (~1.5°C each in the mid-Miocene). The uncertainty on salinity is negligible (<0.1°C) and included here only for the sake of completeness. This information is now given in the text on lines 307-310.

*The evolutionary perspective on differences in Mg/Ca-temperature relationships is very interesting (and slightly annoying for myself: I was just working on something similar…). However, the analysis of offsets in Mg/Ca and del18O versus evolutionary kinships is not making things clearer. Figures 4 and 5 actually suggest that offsets occur here and there, with no single 'branch' containing all the offset results.*

We thank the reviewer for providing the opportunity for us to clarify this point. We analysed multiple species belonging to 18 genera from 8 localities. While it may appear that offsets are here and there on a very broad scale (e.g. it is not the case that lineage-specific Mg/Ca-T relationships only appear in the non-spinose species), systematic offsets from the multispecies calibration is in many cases consistent within a lineage. Specifically, we observe that offsets tend to appear with the origination of a new lineage and to continue to the modern representative(s) of that lineage. While with the current dataset our lineage dependent offset interpretation cannot be conclusively tested, the within-lineage consistency of the offsets that we observe is what you would expect if the appearance of an offset is linked to evolution. In order to make this clearer, we have combined figure 4 and 5, as suggested by the reviewer, and added a summary figure to help visualize the consistent occurrence of the observed offsets.

*It may help to show which lineages fall towards the left-upper side of the average line and which towards the right-lower side (Figure 6). This would mean that three groups can be compared. But with the current figures, it remains very difficult to see if the species with a larger offset from the average (Figure 6) cluster together. I think this needs to be shown more rigorously if the conclusion of this exercise (i.e. that evolution is somehow responsible for the variability in Mg/Ca versus del18O) can hold.*

With the current dataset we cannot conclusively claim that evolution is responsible for the occurrence of offsets and we did not mean to (and in our view do not) phrase it in these terms in the manuscript. However, the consistency of the pattern suggests that this is most likely the case, pending further studies. We have now made this clearer in the text and added a summary figure to better display the offset distribution among lineages (see below).

[Figure]

**Summary figure.** Simplified phylogeny for the offset lineages discussed in the text with the occurrence of morphological changes associated with their evolution highlighted. Colored lines indicate species offset in Mg/Ca in the study dataset relative to a multispecies calibration approach, black lines indicate non-offset species. Red lines indicate offset species with higher Mg/Ca, blue lines offset species with lower Mg/Ca, and purple lines species displaying both types of offsets. Question marks indicate the lack of Mg/Ca data for a given species in the dataset presented here. Phylogeny after Aze et al. (2011) and Spezzaferri et al. (2018). The phylogenetic chart was generated using Mikrotax (Huber et al., 2016; www.mikrotax.org/pforams). The reference time scale in the figure is the Astronomical time scale of Lourens et al. (2004), until base of Chron C6Cn.2n, and Pälike et al. (2006), from top Chron C6Cn.3n until base C13n.

*Also: the black color of some lines in figure 4 and 5 suggest that they are not offset from the average Mg/Ca-del18O, but with the red question mark beside them, we actually don't know, right? This gives the false impression that more species than is maybe the case, fall around the average Mg/Ca-del18O relationship. Can figures 4 and 5 not be combined? If done, it will become clear that the spinose/ non-spinose divide doesn't match the offset/ non-offset divide. This actually makes me most suspicious of the conclusion that there is a link between phylogeny and calcite chemistry (at least on this level).*

We combined figures 4- and 5 and edited them following the reviewer suggestion (see below). We agree it's more complex than spinose/non-spinose as offsets appear in both groups in parallel with the evolution of new lineages. Also, the observed offsets in the different spinose and non-spinose lineages do not necessarily have the same mechanism. We suspect that large structural changes in the text morphology/wall texture (e.g. appearance of *Praeorbulina-Orbulina* or *Pulleniatina*) may be linked to the appearance of an offset through changes in e.g. biomineralization pathways. Vice versa, for *Globigerina* and *Globigerinella*, where test morphology/texture is a lot more conservative through geological time, the occurrence of the offset may be linked to changes in the ecology. We tried and showed this in the new summary figure. However, to conclusively test these interpretations we would need to track the occurrence/absence of offsets down to the Oligocene ancestors of modern/Neogene offset species.

[Figure]

**Combined figure 4&5**. Phylogenetic relationships of offset spinose and non-spinose species. Shown here are the species discussed in the text, their most closely related species and the ancestors. Colored lines indicate species offset in Mg/Ca in the study dataset relative to a multispecies calibration approach, black lines indicate non-offset species. Red lines indicate offset species with higher Mg/Ca, blue lines offset species with lower Mg/Ca, and purple lines species displaying both types of offsets. Question marks indicate the lack of Mg/Ca data for a given species in the dataset presented here. Phylogeny after Aze et al. (2011) and Spezzaferri et al. (2018). The phylogenetic chart was generated using Mikrotax (Huber et al., 2016; www.mikrotax.org/pforams). The reference time scale in the figure is the Astronomical time scale of Lourens et al. (2004), until base of Chron C6Cn.2n, and Pälike et al. (2006), from top Chron C6Cn.3n until base C13n.

*Minor comments:*
*Line 30: remove 'of'*

Ok, done.

*Line 32: The authors argue in the discussion that deviations from the average Mg/Ca-del18O correlation is mainly due to inter-species variability in Mg/Ca (rather than variability in del18O). This may be the case, but interspecies differences in oxygen isotopes may still contribute to the variability shown in figures 6 and 7. Therefore, it may be more accurate here to say 'systematic offsets in the Mg/Ca-del18O relationship' or something similar.*

We fully agree and have edited the text accordingly. Specifically, we have edited the sentence on lines 433-435 for clarity and now also compare the range of inter-species offsets in d18O and Mg/Ca.

*Line 175: I guess you combined the 'Mg/Ca-temperature calibrations'*
Edited accordingly (line 176).

*Lines 205-208: it may be easier to simply state that you corrected the Mg/Cacc for past seawater Mg/Ca and pH (see 2.4.2). I guess it doesn't matter for this procedure what Mg/Ca-temperature calibration you use (since you use it twice in 'opposite' directions).*

The reviewer is correct and we now clarify that this is the case (line 210).

*Line 239: I don't understand this: it seems that the effect of pH, salinity and temperature were taken into account twice. In lines 218-234 it was also described that the long-term changes in salinity and pH were corrected for. Conversion to temperature was already mentioned in line 238.*

We have rephrased for clarity (lines 241-242). The corrections were of course only applied once as the reviewer correctly surmises.

*Line 240: what exponential coefficient? For the G. ruber Mg/Ca-pH sensitivity?*

Here we refer to the pre-exponential coefficient of a regression in the form Mg/Ca = BeAT, which is now clarified on lines 244-245.

*Line 250: what does this re-fit mean precisely? I understood that the Mg/Cacc were already corrected for salinity and pH…*

MgCaRB requires a covariance matrix of parameter uncertainty in order to implement the uncertainty propagation (Gray and Evans, 2019). In order to generate the covariance matrix, we performed a bootstrap of the Tierney et al. (2019) *Neogloboquadrina pachyderma* dataset,

following the method outlined in Gray and Evans (2019), prescribing the calibration sensitivities and uncertainties from Tierney et al 2019. The calibration dataset and uncertainty covariance are shown in figure X. As a sensitivity experiment we performed the same analysis, however prescribing the 'generic' pH sensitivity of Gray and Evans (2019).

[Figure]

**We propose to add the figure above as supplementary figure: (a)** *Neogloboquadrina pachyderma* calibration dataset of Tierney et al 2019 (showing the temperature sensitivity determined in that study) and **(b)** covariance between temperature sensitivity and the 'intercept' term implemented in MgCaRB.

*Figure 2: so, since the different corrections are plotted 'on top' of each other, the order in which they are stacked matters for the suggestion that only part of the data could be accounted for with these corrections. The order could also be such that the upper left part of the data cloud falls within the calibration lines. It is not important for the paper, but perhaps the authors could try to change this? The applied corrections are actually vectors pushing the original calibration in different directions…*

We agree with the reviewer that this could be a more intuitive way of displaying the relative importance of the various nonthermal controls on the proxies, and we did not do so only because in several cases it is not possible - some effects cannot be described as vectors in Mg/Ca-d18O space. For example, pH impacts both Mg/Ca and d18O and has the effect of rotating the slope of the relationship between the two proxies when they are plotted against each other. That is, the slope of the line becomes shallower and the vector applicable to the bottom-left of the diagram would not be the same as that applicable to the top-right.

*Figure 2: I don't understand the Mg/Casw =2.5. That is (almost) half that of today, but in the methods there lacks a justification for this. Instead, the methods describe how the reconstructed change in Mg/Ca is used to correct the Mg/Cacc. It makes me wonder in figure 2, then what this line actually signifies: the caption suggests that the change in Mg/Casw from 15 Ma till now was used to correct the Mg/Ca of the foraminifera, but this would not (necessarily) result in a constant offset (as the orange line here suggests).*

The logic for choosing Mg/CaSW = ~50% present day is that this is the lowest possible value that could apply to a sample presented here, that is, it is roughly equivalent to the lower uncertainty bound on early-middle Miocene Mg/CaSW. We chose the largest possible change to make the effect easier to see and do not mean to apply that this is the value that is applicable to any specific sample - the idea of this figure in general is to visualise the sensitivity of the proxy values to the various controls. This is now clarified in the figure caption (lines 288-292).

*Figure 2B: what does the 'm = -2.08' mean?*
We agree that the meaning of this line was not clear, and have now clarified in the figure caption (lines 296-301).

*Line 284-285: 'out of' should be 'from'.*
Now corrected.

*Line 311: I tried to look it up, but couldn't find how many datapoints were generated in total. The 62% sounds a higher percentage than it looks to me, but there may simply be many datapoints obscured by the density of the center of the cloud.*
Yes, they all cluster together, all data are available in the supplementary table.

*Line 352: should be 'addition'*
Now corrected.

*Line 434 and on: what do the ratios between parentheses mean?*
Now clarified.

*Line 442: reference to Fig 5 is incorrect here. Also, references to figure 5 in the text of the manuscript appear after references to figure 6.*

We double checked this and the references to the figures appear correct to us. Figure 5 is mentioned for the first time in line 436, Figure 6 in line 439, so this also looks correct. In any case, now the figure numbering in the text has changed as a result of the inclusion of a new figure and all call outs to figures have been updated accordingly.

*Figure 4: what are 'offset' spinose species? If it refers to species that have a del18O-MgCa relationship that deviates too much from the average relationship (figure 6), then what exactly makes it offset? What is the threshold value for such a distinction and what is this based on?*

We define a species offset when it plots outside the combined error envelope of the two proxies, hence when the difference between the reconstructed temperature from the two proxies is >~5°C (although this is evaluated on a case-by-case basis using the specific uncertainties for a given data point). This is described in lines 266-268 in the Methods and lines 498-501 in the Discussion. We also now explain this in the figure captions of Figure 6 and 7.

*Line 539: should be 'as in Figure 6'.*
Now corrected.

---

## Author Comment (AC2)

**Review rebuttal for:**

**Mg/Ca and δ¹⁸O in multiple species of planktonic foraminifera from 15 Ma to Recent**

By Boscolo-Galazzo et al.

We thank the reviewers and editor for their comments and feedback on the manuscript. We agree with almost all of the points raised and have revised the text accordingly. Below we give a point-by-point response to the reviewers comments.

**Reviewer 2**

*d18O-temperatures: I'm aware that the online tool of Gaskell and Hull (2023) allows for a limited number of d18Osw corrections and that Rohling et al. (2021) (R_21) is a perfectly valid option. However, not only is d18Osw beyond the Pleistocene (or even just the LGM) quite uncertain, but also a recent benthic d18O deconvolution from Rohling et al. (2022) (R_22 https://doi.org/10.1029/2022RG000775) suggests heavier d18Osw during most of the last 15 million years compared to R_21, with the independent model output from De Boer et al. (2010) sitting somewhere in between R_21 and R_22. Meanwhile, clumped isotope derived d18Osw suggests even heavier values for parts of the Miocene (e.g. https://doi.org/10.1029/2020PA003927). For these reasons, I wonder if it'd be possible to at least account for some of this uncertainty in your d18O-temperatures (larger vertical error bars?) and whether this might bring some of your "discordant" samples back into the uncertainty envelope of Figure 3.*

We thank the reviewer for raising this important point.
As the reviewer points out, using an alternative sea level/deep ocean temperature record would potentially systematically shift the d18O-temperature reconstructions. For example, using the reconstructions of Miller et al. (2020) rather than Rohling et al. (2022) would result in d18O-derived temperatures 0.6-3.3°C warmer than our preferred scenario as presented in the main text, with the difference most pronounced for our 7.5 Ma samples, changing the overall proxy-proxy agreement from 62% to 64%. However, rather than include this source of uncertainty in the error bars of individual data points, we have accommodated this suggestion by including a supplementary version of main text Fig. 3 alternatively using Miller et al. (2020). The rationale for this is that, as the reviewer notes, many reconstructions are available, each with underlying assumptions or issues and it is not clear that all should be given equal weight in terms of application to paleo reconstructions. For example, in this specific case, we opt for the use of Rohling et al. (2022) over Miller et al. (2020) due to differing treatment of the benthic foraminifera Mg/Ca data used to deconvolve the benthic d18O record. However, given that our manuscript does not present any new benthic data and therefore cannot contribute directly to this debate, we do not feel it is the right place to critique individual approaches. Rather, we stress in the manuscript that these are the temperatures one would arrive at for a given underlying set of

assumptions, which are clearly noted, but that alternate choices have the potential to systematically shift the d18O temperature data in particular (lines 331-340). Encouragingly, while the choice of sea level record mentioned above can impact d18O temperature by up to ~3°C this has very little impact on our interpretation overall as the dataset is approximately equally distributed either side of the 1:1 line.

[Figure]

**We propose to add the figure above as supplementary figure:** it shows main text Fig. 3 except using the sea level and deep ocean temperature record of Miller et al. (2020) to derive d18Osw following the approach of Gaskell et al. (2023).

*Alternatively, you could replot a few more versions of Figure 3 with different d18Osw corrections as offered in Gaskell and Hull (2023). Either way, I think it would be important to at least acknowledge these uncertainties in d18Osw before giving the reader the impression that a high degree of confidence exists behind the non-thermal corrections mentioned in the text. Moreover, this has implications for e.g. your hypothesis in Section 4.1.*

Please see our response to the previous comment. We have done exactly what the reviewer suggests here (proposed supplementary figure above).

*Technical corrections*

*Title: The title is too general, you could consider making it a bit more specific and refer to some of your main results.*

We respectfully disagree with the reviewer on this point. In our view, the title accurately reflects what the study is about and, while we appreciate that it is often helpful to place a key result in the

title, it is difficult to do so in this case given that we demonstrate the existence of lineage-specific effects on Mg/Ca.

*(Line) 220: The sea level record of Spratt and Liesecki only reaches 800 kyrs and doesn't go as far as 8 Ma as stated on line 221.*

The reviewer is correct, we simply use Spratt & Liesecki to derive a scaling factor that is then applied back to 8 Ma. We have rephrased for clarity (line 223).

*Also, I understand that in order to use the tool of Gray and Evans (2019) in samples older than 800 kyrs you probably had to modify the original code to include longer datasets for the salinity and and pH corrections (I only have experience with the R version so I'm not sure if the matlab version is a bit more flexible in this sense). If that were the case you could be more explicit with this in the text.*

Yes, we did update the code. Upon acceptance we will place the revised version of the script on the MgCaRB github page (https://github.com/dbjevans/MgCaRB) and add a note to the text will be added.

*221: Given that your choice of sea level/ benthic d18O deconvolution is Rohling et al., 2022 for the conversion of d18O into temperature (Line 259) I suggest using this same sea level record instead of Miller et al., 2005 for this step as well (from 800 kyrs up to 15 Ma, see comment for Line 220) to be as consistent as possible with the corrections applied to both proxies.*

Yes, we agree. This has now been updated in the code and the figures changed but it has virtually no impact on the results as the salinity control on Mg/Ca is extremely minor.

*229: This line suggests a very similar (or equivalent?) Mg/Casw than that one derived in Rosenthal, Bova and Zhou (2022) where they also combined Zhou et al., 2021 with the fluid inclusion data from Brennan et al., 2013. If it is then it might be more clear for the reader to cite the Rosenthal and Bova (2022) record directly (https://www.ncei.noaa.gov/access/paleo-search/study/36413).*

Yes, we now cite this as suggested.

*Figures 3B-8B: These plots colored by calcification depth are so interesting it would be nice if you could increase the color contrast a bit more so that the 200-0 m range is more visible.*

Thank you, we tried and made it stand out more.

*Line 626: It goes from ~4 mol/mol at 5 Ma to ~5.25 mol/mol at present (I'm looking at Figure 10 in Rosenthal, Bova and Zhou, 2022). did you perhaps mean 15 Ma?*

Yes, corrected and reference to Rosenthal et al. (2022) added.

---

## Author Response (AR1)

Editor Decision

Dear Flavia Boscolo-Galazzo and co-authors,

Your manuscript has received two review reports and after reading through them and your Author Comment, I have decided that the moderate revisions you suggest to do to your manuscript will likely answer the concerns raised by the reviewers. I therefore invite you to make these revisions and resubmit a revised version of your manuscript. After resubmission, you can expect me to ask at least one of the reviewers for a second opinion. During your revision, I suggest you pay specific attention to the following points raised by the reviewers:

Firstly, the statistical approach underlying your data analysis needs to be clear for the reader of Biogeosciences, meaning that readers not familiar with Monte Carlo simulations and error propagation need to be able to follow your workflow. From the comment of Reviewer #1 it seems this requires some more attention.

Secondly, the discussion on how offsets from the Mg/Ca-d18O relationship maps on phylogeny could probably be sharpened a bit. From the Abstract and the Discussion, it should be clear what can and cannot be concluded about evolutionary control on Mg/Ca incorporation in foraminifera based on this dataset. The merged Figure 4&5 combination will help with that, but please ensure that it is also clear from your text what conclusions can be drawn from this figure.

Thirdly, I believe Reviewer #2 raises a good point about the title of your work. Since, according to your Author Reply, the main conclusion of the study is the existence of lineage-specific effects on Mg/Ca, I believe adding this notion to the title would make it clearer for the reader what can be expected from the paper.

Fourthly, in the spirit of open science, please already refer to Github code in your next revised version to give reviewer access to the code you used to process the data. Making this accessible only on acceptance in theory prevents the reviewer from checking on this fundamental part of your work. As Github is a versioned database, I additionally suggest that you create a release of your Github repository to cite (e.g. through Zenodo link) and that you refer to this release in the next revised version of your manuscript. Open science is a core value of the Copernicus journals and I will not accept a manuscript for publication before the code used in its methodology is openly available.

Finally, I think the main point by Reviewer #2 about the d18O value of the ocean water is important, but well addressed in the author comment. I agree with you that the choice of d18Osw model should not necessarily feature as part of the uncertainty in your data analysis. However, please make sure that, throughout the manuscript, it is clear how your discussion and conclusion depends on the choice of d18Osw model. The suggested supplementary figure is a good start.

Thank you again for submitting your work to Biogeosciences and I look forward to receiving your revised manuscript.

Kind regards, Niels de Winter

Reply to the Editor and Reviewers

Dear Editor,

Thank you very much for your positive evaluation of our manuscript and our proposed revisions. We paid great attention to carefully address yours and the reviewers' suggestions and changed the manuscript accordingly. We think the changes we made to the manuscript fully satisfy the requested revisions.

Specifically,

*R1: The statistical approaches should be explicitly described. This includes the exact way the regression analyses were done and the way the MC simulation was applied. Please provide the necessary details. Line 254: do the 'calibration coefficients' refer to the uncertainty in the estimated constants from the regression analysis? It would be interesting to know which of the sources of the uncertainty contribute most to the overall CI.*

To clarify the text, we have rephrased the part which describes the Monte Carlo simulations specifically (lines 267-273). Further, we now give more details of uncertainty propagation and the sources of uncertainty that are propagated in the Supplement (Supplementary Figure 3). Specifically, the main sources of uncertainties are derived from: the calibration coefficients, Mg/Ca$_{sw}$, pH, and in the case of d18O, the d18O$_{sw}$. For Mg/Ca, the magnitude of these uncertainties is approximately split between the uncertainty on the calibration coefficients and the uncertainty on the Mg/Ca$_{sw}$ reconstruction (~1.5°C each in the mid-Miocene). The uncertainty on salinity is negligible (<0.1°C) and included here only for the sake of completeness.

The Monte Carlo error propagation itself is performed and described in a standard way in the text, in our view sufficient information is given here to replicate our calculations, but we would of course be happy to add any further specific details the reviewer has in mind.

*R1: The evolutionary perspective on differences in Mg/Ca-temperature relationships is very interesting (and slightly annoying for myself: I was just working on something similar…). However, the analysis of offsets in Mg/Ca and del18O versus evolutionary kinships is not making things clearer. Figures 4 and 5 actually suggest that offsets occur here and there, with no single 'branch' containing all the offset results.*

We thank the reviewer for providing the opportunity for us to clarify this point. We analysed multiple species belonging to 18 genera from 8 localities. While it may appear that offsets are here and there on a very broad scale (e.g., it is not the case that lineage-specific Mg/Ca-T relationships only appear in the non-spinose species), systematic offsets from the multispecies calibration is in many cases consistent within a lineage. Specifically, we observe that offsets tend to appear with the origination of a new lineage and to continue to the modern representative(s) of that lineage. While with the current dataset our lineage dependent offset interpretation cannot be conclusively tested, the within-lineage consistency of the offsets that we observe is what you would expect if the appearance of an offset is linked to evolution. In order to make this clearer, we have combined figure 4 and 5, as suggested by the reviewer, and added a summary figure to help visualize the consistent occurrence of the observed offsets.

R1: *It may help to show which lineages fall towards the left-upper side of the average line and which towards the right-lower side (Figure 6). This would mean that three groups can be compared. But with the current figures, it remains very difficult to see if the species with a larger offset from the average (Figure 6) cluster together. I think this needs to be shown more rigorously if the conclusion of this exercise (i.e., that evolution is somehow responsible for the variability in Mg/Ca versus del18O) can hold.*

With the current dataset we cannot conclusively claim that evolution is responsible for the occurrence of offsets and we did not mean to (and in our view do not) phrase it in these terms in the manuscript. However, the consistency of offset distribution between lineages as well as the improvement in temperature reconstructions when lineage specific calibrations are applied, suggest that this is most likely the case. We have now made clearer in the text (Discussion and Conclusion) that ours is a resonate and reasonable suggestion but still a suggestion, and added a summary figure (Figure 8) to better display the offset distribution among lineages. We think this is the most honest, reasonable and transparent interpretation of the data and data analysis that we can provide.

R2: *Also: the black color of some lines in figure 4 and 5 suggest that they are not offset from the average Mg/Ca-del18O, but with the red question mark beside them, we actually don't know, right? This gives the false impression that more species than is maybe the case, fall around the average Mg/Ca-del18O relationship. Can figures 4 and 5 not be combined? If done, it will become clear that the spinose/ non-spinose divide doesn't match the offset/ non-offset divide. This actually makes me most suspicious of the conclusion that there is a link between phylogeny and calcite chemistry (at least on this level).*

We combined figures 4 and 5 and edited them following the reviewer suggestion. We agree it's more complex than spinose/non-spinose as offsets appear in both groups in parallel with the evolution of new lineages. Also, the observed offsets in the different spinose and non-spinose lineages do not necessarily have the same mechanism. We suspect that large structural changes in the text morphology/wall texture (e.g., appearance of Praeorbulina-Orbulina or Pulleniatina) may be linked to the appearance of an offset through changes in e.g., biomineralization pathways. Viceversa, for Globigerina and Globigerinella, where test morphology/texture is a lot more conservative through geological time, the occurrence of the offset may be linked to changes in the ecology, e.g., because of the appearance of new genotypes in these particularly genotype-rich species. We tried and showed this in the new summary figure (Figure 8). However, to conclusively test these interpretations we would need to track the occurrence/absence of offsets down to the Oligocene ancestors of modern/Neogene species.

R1: *Minor comments:*

*Line 30: remove 'of'*

Ok, done.

*Line 32: The authors argue in the discussion that deviations from the average Mg/Ca-del18O correlation is mainly due to inter-species variability in Mg/Ca (rather than variability in del18O). This may be the case, but interspecies differences in oxygen isotopes may still contribute to the variability shown in figures 6 and 7. Therefore, it*

*may be more accurate here to say 'systematic offsets in the Mg/Ca-del18O relationship' or something similar.*

We fully agree and have edited the text accordingly. Specifically, we have edited the sentence on lines 437-439 for clarity and now also compare the range of inter-species offsets in d18O and Mg/Ca.

*Line 175: I guess you combined the 'Mg/Ca-temperature calibrations'*

Edited accordingly (line 165).

*Lines 205-208: it may be easier to simply state that you corrected the Mg/Cacc for past seawater Mg/Ca and pH (see 2.4.2). I guess it doesn't matter for this procedure what Mg/Ca-temperature calibration you use (since you use it twice in 'opposite' directions).*

The reviewer is correct and we now clarify that this is the case (line 199).

*Line 239: I don't understand this: it seems that the effect of pH, salinity and temperature were taken into account twice. In lines 218-234 it was also described that the long-term changes in salinity and pH were corrected for. Conversion to temperature was already mentioned in line 238.*

We have rephrased the whole paragraph for clarity (lines 248-255). The corrections were of course only applied once as the reviewer correctly surmises.

*Line 240: what exponential coefficient? For the G. ruber Mg/Ca-pH sensitivity?*

Here we refer to the pre-exponential coefficient of a regression in the form Mg/Ca = BeAT, which is now clarified on lines 252-253.

*Line 250: what does this re-fit mean precisely? I understood that the Mg/Cacc were already corrected for salinity and pH…*

MgCaRB requires a covariance matrix of parameter uncertainty in order to implement the uncertainty propagation (Gray and Evans, 2019). In order to generate the covariance matrix, we performed a bootstrap of the Tierney et al. (2019) Neogloboquadrina pachyderma dataset, following the method outlined in Gray and Evans (2019), prescribing the calibration sensitivities and uncertainties from Tierney et al 2019. This allows us to implement the N. pachyderma calibration of Tierney et al (2019) within the MgCaRB script. The calibration dataset and uncertainty covariance are shown in Supplementary Figure 1. As a sensitivity experiment, we performed the same analysis, however prescribing the 'generic' pH sensitivity of Gray and Evans (2019).

*Figure 2: so, since the different corrections are plotted 'on top' of each other, the order in which they are stacked matters for the suggestion that only part of the data could be accounted for with these corrections. The order could also be such that the upper left part of the data cloud falls within the calibration lines. It is not important for the paper, but perhaps the authors could try to change this? The applied corrections are actually vectors pushing the original calibration in different directions…*

We agree with the reviewer that this could be a more intuitive way of displaying the relative importance of the various nonthermal controls on the proxies, and we did not do so only because in several cases it is not possible - some effects cannot be described as vectors in Mg/Ca-d18O space. For example, pH impacts both Mg/Ca and d18O and has the effect of rotating the slope of the relationship between the two proxies when they are plotted against each other. That is, the slope of the line becomes

shallower and the vector applicable to the bottom-left of the diagram would not be the same as that applicable to the top-right.

*Figure 2: I don't understand the Mg/Casw =2.5. That is (almost) half that of today, but in the methods there lacks a justification for this. Instead, the methods describe how the reconstructed change in Mg/Ca is used to correct the Mg/Cacc. It makes me wonder in figure 2, then what this line actually signifies: the caption suggests that the change in Mg/Casw from 15 Ma till now was used to correct the Mg/Ca of the foraminifera, but this would not (necessarily) result in a constant offset (as the orange line here suggests).*

The logic for choosing Mg/Casw = ~50% present day is that this is the lowest possible value that could apply to a sample presented here, that is, it is roughly equivalent to the lower uncertainty bound on early-middle Miocene Mg/Casw. We chose the largest possible change to make the effect easier to see and do not mean to apply that this is the value that is applicable to any specific sample - the idea of this figure in general is to visualise the sensitivity of the proxy values to the various controls. This is now clarified in the figure caption (lines 211-214).

*Figure 2B: what does the 'm = -2.08' mean?*

We agree that the meaning of this line was not clear, and have now clarified in the figure caption (lines 219-223).

*Line 284-285: 'out of' should be 'from'.*

Now corrected.

*Line 311: I tried to look it up, but couldn't find how many datapoints were generated in total. The 62% sounds a higher percentage than it looks to me, but there may simply be many datapoints obscured by the density of the center of the cloud.*

Yes, they all cluster together, all data are available in the supplementary table.

*Line 352: should be 'addition'.*

Now corrected.

*Line 434 and on: what do the ratios between parentheses mean?*

Now clarified.

*Line 442: reference to Fig 5 is incorrect here. Also, references to figure 5 in the text of the manuscript appear after references to figure 6.*

We double checked this and the references to the figures appear correct to us. Figure 5 is mentioned for the first time in line 436, Figure 6 in line 439, so this also looks correct. In any case, now the figure numbering in the text has changed as a result of the inclusion of a new figure and all call outs to figures have been updated accordingly.

*Figure 4: what are 'offset' spinose species? If it refers to species that have a del18O-MgCa relationship that deviates too much from the average relationship (figure 6), then what exactly makes it offset? What is the threshold value for such a distinction and what is this based on?*

Offsets are initially qualitatively described as Mg/Ca signatures which do not match the corresponding d18O signatures either because they are too low (non-spinose species) or too high (spinose) or both (Globigerinella) (lines 435-456), for instance compare

Fig.6 a, c (non offset species) with Fig. 6 e, g, i, k (offset species). When we convert the data into temperature, we observe that the species presenting offset Mg/Ca values also show offset Mg/Ca temperatures (Fig. 6 f, h, j, l) (Supplementary Table 1). We hence use the combined error envelope of the two proxies as an objective divide to quantitatively distinguish between offset and non-offset species. Specifically, we define a species offset when it plots outside the combined error envelope of the two proxies, hence when the difference between the reconstructed temperature from the two proxies is >~5°C (although this is evaluated on a case-by-case basis using the specific uncertainties for a given data point). This is described in lines 281-283 in the Methods and lines 513-522 in the Discussion. We also now explain this in the figure captions of Figure 5 and 6.

*Line 539: should be 'as in Figure 6'.*

Now corrected.

*R2: d18O-temperatures: I'm aware that the online tool of Gaskell and Hull (2023) allows for a limited number of d18Osw corrections and that Rohling et al. (2021) (R_21) is a perfectly valid option. However, not only is d18Osw beyond the Pleistocene (or even just the LGM) quite uncertain, but also a recent benthic d18O deconvolution from Rohling et al. (2022) (R_22 https://doi.org/10.1029/2022RG000775) suggests heavier d18Osw during most of the last 15 million years compared to R_21, with the independent model output from De Boer et al. (2010) sitting somewhere in between R_21 and R_22. Meanwhile, clumped isotope derived d18Osw suggests even heavier values for parts of the Miocene (e.g.https://doi.org/10.1029/2020PA003927). For these reasons, I wonder if it'd be possible to at least account for some of this uncertainty in your d18O-temperatures (larger vertical error bars?) and whether this might bring some of your "discordant" samples back into the uncertainty envelope of Figure 3.*

We thank the reviewer for raising this important point. As the reviewer points out, using an alternative sea level/deep ocean temperature record would potentially systematically shift the d18O-temperature reconstructions. For example, using the reconstructions of Miller et al. (2020) rather than Rohling et al. (2022) would result in d18O-derived temperatures 0.6-3.3°C warmer than our preferred scenario as presented in the main text, with the difference most pronounced for our 7.5 Ma samples, changing the overall proxy-proxy agreement from 62% to 64%. However, rather than include this source of uncertainty in the error bars of individual data points, we have accommodated this suggestion by including a supplementary version of main text Fig. 3 alternatively using Miller et al. (2020) (Supplementary Figure 2). The rationale for this is that, as the reviewer notes, many reconstructions are available, each with underlying assumptions or issues and it is not clear that all should be given equal weight in terms of application to paleo reconstructions. For example, in this specific case, we opt for the use of Rohling et al. (2022) over Miller et al. (2020) due to differing treatment of the benthic foraminifera Mg/Ca data used to deconvolve the benthic d18O record. However, given that our manuscript does not present any new benthic data and therefore cannot contribute directly to this debate, we do not feel it is the right place to critique individual approaches. Rather, we stress in the manuscript that these are the temperatures one would arrive at for a given underlying set of assumptions, which are clearly noted, but that alternate choices have the potential to systematically shift the d18O temperature data in particular (lines 315-321). Encouragingly, while the choice of sea level record mentioned above can impact d18O temperature by up to ~3°C this has very little impact on our interpretation overall as the dataset is approximately equally distributed either side of the 1:1 line.

*R2: Alternatively, you could replot a few more versions of Figure 3 with different d18Osw corrections as offered in Gaskell and Hull (2023). Either way, I think it would be important to at least acknowledge these uncertainties in d18Osw before giving the reader the impression that a high degree of confidence exists behind the non-thermal corrections mentioned in the text. Moreover, this has implications for e.g., your hypothesis in Section 4.1.*

Please see our response to the previous comment. We have done exactly what the reviewer suggests here (Supplementary Figure 2).

R2: Technical corrections

*Title: The title is too general, you could consider making it a bit more specific and refer to some of your main results.*

We try and provide a new version of the title which refers to our macroevolutionary interpretation of our results. However, we would like to stress that the dataset we analyse here is that of Boscolo-Galazzo et al. (2021) which was not obtained with the intent of testing for evolutionary fingerprints in foraminiferal shell elemental chemistry. This was rather something that came up when we started to compare the species-specific d18O-Mg/Ca for this work as well as with our data analysis. Hence, while we think that the occurrence of lineage specific offsets in Mg/Ca is a new and noteworthy finding of this study which requires to be highlighted and given the right importance, we do not have the means (data) to conclusively test its relationship with evolution. We can only provide this interpretation to the best of our knowledge and "open the floor" for discussion and further investigation within the community.

*(Line) 220: The sea level record of Spratt and Liesecki only reaches 800 kyrs and doesn't go as far as 8 Ma as stated on line 221.*

The reviewer is correct, we simply use Spratt & Liesecki to derive a scaling factor that is then applied back to 8 Ma. We have rephrased for clarity (line 234).

*Also, I understand that in order to use the tool of Gray and Evans (2019) in samples older than 800 kyrs you probably had to modify the original code to include longer datasets for the salinity and pH corrections (I only have experience with the R version so I'm not sure if the matlab version is a bit more flexible in this sense). If that were the case you could be more explicit with this in the text.*

Yes, we did update the code. The updated code can be found here: https://github.com/dbjevans/MgCaRB/releases/tag/v1.3

*221: Given that your choice of sea level/ benthic d18O deconvolution is Rohling et al., 2022 for the conversion of d18O into temperature (Line 259) I suggest using this same sea level record instead of Miller et al., 2005 for this step as well (from 800 kyrs up to 15 Ma, see comment for Line 220) to be as consistent as possible with the corrections applied to both proxies.*

Yes, we agree. This has now been updated in the code and the figures changed (Line 235) but it has virtually no impact on the results as the salinity control on Mg/Ca is extremely minor.

*229: This line suggests a very similar (or equivalent?) Mg/Casw than that one derived in Rosenthal, Bova and Zhou (2022) where they also combined Zhou et al., 2021 with the fluid inclusion data from Brennan et al., 2013. If it is then it might be more clear for*

*the reader to cite the Rosenthal and Bova (2022) record directly (https://www.ncei.noaa.gov/access/paleo-search/study/36413).*

Yes, we now cite this as suggested.

*Figures 3B-8B: These plots colored by calcification depth are so interesting it would be nice if you could increase the color contrast a bit more so that the 200-0 m range is more visible.*

Thank you, we tried and made it stand out more.

*Line 626: It goes from ~4 mol/mol at 5 Ma to ~5.25 mol/mol at present (I'm looking at Figure 10 in Rosenthal, Bova and Zhou, 2022). did you perhaps mean 15 Ma?*

Yes, corrected and reference to Rosenthal et al. (2022) added.